# Ion transfer mechanisms in Mrp-type antiporters from high resolution cryoEM and molecular dynamics simulations

Yongchan Lee [1,6,7], Outi Haapanen [2,7], Anton Altmeyer [3,4], Werner Kühlbrandt [1], Vivek Sharma [2,5] ✉ & Volker Zickermann [3,4] ✉

Multiple resistance and pH adaptation (Mrp) cation/proton antiporters are essential for growth of a variety of halophilic and alkaliphilic bacteria under stress conditions. Mrp-type antiporters are closely related to the membrane domain of respiratory complex I. We determined the structure of the Mrp antiporter from *Bacillus pseudofirmus* by electron cryo-microscopy at 2.2 Å resolution. The structure resolves more than 99% of the sidechains of the seven membrane subunits MrpA to MrpG plus 360 water molecules, including ~70 in putative ion translocation pathways. Molecular dynamics simulations based on the high-resolution structure revealed details of the antiport mechanism. We find that switching the position of a histidine residue between three hydrated pathways in the MrpA subunit is critical for proton transfer that drives gated trans-membrane sodium translocation. Several lines of evidence indicate that the same histidine-switch mechanism operates in respiratory complex I.

Sodium/proton antiporters are essential to maintain an intracellular homeostasis of pH and sodium concentration[1-3]. Cation/proton antiporters (CPAs) of the CPA1 and CPA2 families, such as NhaA and NHE, are structurally and functionally fully characterized[2,4]. The CPA3 family comprises the larger Mrp (multiple resistance and pH adaptation) type antiporters which have received much attention because of their evolutionary relationship to redox-driven ion pumps of the complex I superfamily[5-9]. Energy-converting complexes of the complex I type all combine different sets of Mrp antiporter-like membrane subunits with a membrane-anchored hydrophilic redox unit.

The Mrp operon was first identified in a *Bacillus* species[10,11]. Since then, Mrp-type antiporters have been found in many halophilic and alkaliphilic bacteria[12]. They can be divided into two large groups. Group I Mrp antiporters consist of seven proteins (MrpA-MrpG). In group II, MrpA and B are fused, while all other proteins are single polypeptides (MrpA´ and MrpC-MrpG). The hetero-oligomeric complexes have been isolated in monomeric and dimeric form[13-15]. CryoEM structures of two Mrp antiporters have been determined recently, one from *Dietzia* sp.[15], and the other from *Anoxybacillus flavithermus*[14]. Both differ considerably from those of the other two CPA families.

The largest subunit MrpA consists of two domains connected by a helix on the cytoplasmic side, sometimes referred to as the lateral helix. The overall structure of the N-terminal core domain is similar to that of the MrpD subunit, and the fold is also conserved in respiratory complex I subunits ND5, ND4, and ND2[14-16]. The C-terminal domain of MrpA resembles the ND6 subunit of complex I[17]. MrpC is related to complex I subunit ND4L[17]. More recently, it was shown that also subunits of membrane-bound hydrogenase (MBH)[18] and elemental sulfur reductase (MBS)[19] are clearly related to subunits of the antiporter.

[1]Department of Structural Biology, Max Planck Institute of Biophysics, 60438 Frankfurt am Main, Germany. [2]Department of Physics, University of Helsinki, 00014 Helsinki, Finland. [3]Institute of Biochemistry II, University Hospital, Goethe University, 60590 Frankfurt am Main, Germany. [4]Centre for Biomolecular Magnetic Resonance, Institute of Biophysical Chemistry, Goethe University, 60438 Frankfurt am Main, Germany. [5]HiLIFE Institute of Biotechnology, University of Helsinki, 00014 Helsinki, Finland. [6]Present address: Graduate School of Medical Life Science, Yokohama City University, 230-0045 Kanagawa, Japan. [7]These authors contributed equally: Yongchan Lee, Outi Haapanen. ✉e-mail: vivek.sharma@helsinki.fi; zickermann@med.uni-frankfurt.de

In the single-subunit antiporter NhaA, a few closely spaced residues in the center of the protein have a key function for binding of sodium ions or protons, and the transported ions move along the same funnel-like pathways to either side of the membrane[4]. In contrast, the ion translocation pathways in the multi-subunit protein complexes of the complex I superfamily and in Mrp-type antiporters are debated. For the antiporter, several mutually exclusive models have been suggested[5,14,15,18]. Steiner and Sazanov propose that trans-membrane proton transfer is carried out by the MrpA and MrpD subunits while sodium is taken up from the cytoplasm by MrpE and released to the periplasm at the interface of MrpA, MrpC, and MrpD[14]. In contrast, Li et al.[15] assign proton translocation to MrpA and sodium translocation to MrpD with an exit pathway similar to that proposed independently in ref. [14]. Based on molecular simulations, a recently published study[20] suggests transfer pathways for sodium in MBH that differ significantly from the original interpretations of the structure[18]. With respect to complex I, we and others have recently questioned the previous consensus model that assumes four complete and self-contained proton pumping paths[21,22]. A molecular understanding of the Mrp antiporter is therefore also of great importance for fundamental questions relating to complex I function and biological energy conversion.

In this work, we determine the cryoEM structure of the Mrp antiporter from *Bacillus pseudofirmus* at 2.2 Å resolution. The antiporter complex from this organism is well studied biochemically and by site-directed mutagenesis[13,23–25]. Our high-resolution structure enables us to identify ion translocation pathways by modeling water molecules in hydrophobic trans-membrane regions. Our maps give clear indications for conformational changes in key regions of the structure with far-reaching mechanistic implications. Molecular dynamics (MD) simulations performed on the basis of the high-resolution cryoEM structure provide the atomistic determinants underlying observed conformational changes. In addition, MD simulations reveal putative sodium ion binding sites and how sodium ion translocation may depend on protonation and conformational states of the antiporter.

## Results

### High resolution structure

We expressed recombinant His-tagged *B. pseudofirmus* Mrp antiporter complex in *E. coli* and purified it by affinity chromatography in lauryl maltose neopentyl glycol (LMNG). Electron cryo-microscopy (cryoEM) of the purified complex revealed a dimer of hetero-heptamers, consistent with previous blue-native PAGE analyses of this particular complex[13]. Single-particle data analysis showed varying angles between the two protomers, limiting the map resolution to around 3 Å even after extensive 3D classification. Therefore, we resorted to symmetry expansion and focused 3D refinement (Supplementary Fig. 1, Supplementary Table 1), in an approach similar to that taken by Steiner and Sazanov[14] for the ortholog Mrp antiporter from *A. flavithermus*. The final 3D reconstruction of the protomer yielded a 2.2-Å map, allowing us to build a model including 1957 amino acid residues, three phospholipids, and 360 water molecules (Supplementary Fig. 2, Supplementary Table 2). The detergent belt in the dimer map presented in Fig. 1a gives an indication of the approximate position and shape of the membrane bilayer and indicates a thinning of the membrane in the region of the dimerization domain as previously observed for the *A. flavithermus* Mrp antiporter[14]. The map did not show significant differences between the two protomers in the dimeric complex, and therefore we can confine ourselves to a discussion of the monomeric structure.

Each protomer of the dimeric complex has a mass of 213 kDa and comprises seven subunits that are designated MrpA-G (Fig. 1). The fold of the N-terminal domain of MrpA with 14 trans-membrane helices (TMH1-14) is very similar to that of the neighboring MrpD subunit. In MrpA, this core structure is extended by the so-called lateral helix that runs along the cytoplasmic membrane surface and is anchored by two TMHs, one at the distal end of MrpA and one close to the MrpD subunit. Each of the small subunits MrpC, MrpF, and MrpG consists of a linear arrangement of three TMH each. The C-terminal domain of MrpA (TMH 17-21) surrounds MrpC in a semicircle and contributes a layer of TMHs between MrpC and MrpF. The successive arrangement of MrpC, the C-terminal MrpA domain, MrpF, and MrpG is lined by the four TMHs of the MrpB subunit. MrpE completes the structure and consists of two remarkably short TMHs and a cytoplasmic domain. MrpE forms contacts with MrpE and MrpB from the other protomer. The presence of two short TMHs and horizontal helices in MrpE might cause a thinning of the membrane at the dimer interface as suggested for the Mrp antiporter of *A. flavithermus*[14] (see more below).

The similarity of subunits of the Mrp antiporter with subunits of complex I and members of the complex I superfamily[14–17,26] is summarized in Supplementary Table 3. Note that the conservation of amino acid residues thought to be functionally significant is extensive but not complete (Supplementary Figs. 3 and 4, Supplementary Table 4).

The overall structure of the Mrp antiporter from *B. pseudofirmus* agrees well with the recently published 2.98 Å structure of the antiporter dimer from *A. flavithermus*[14] (Supplementary Fig. 5), but there are obvious differences from the 3.0 Å structure of the monomeric Mrp antiporter from *Dietzia* sp.[15], which belongs to group 2. In the *Dietzia* sp. antiporter, MrpA and MrpB are fused and the dimerization domain in MrpE is absent. All three structures show slight differences in position and inclination of the TMHs of MrpF and MrpG. Our structure is of considerably higher resolution, which enabled us to model protein-bound water molecules (Fig. 2, Supplementary Fig. 6). Furthermore, our map provides evidence of conformational changes that are most likely central to the transport cycle of the antiporter mechanism (see below).

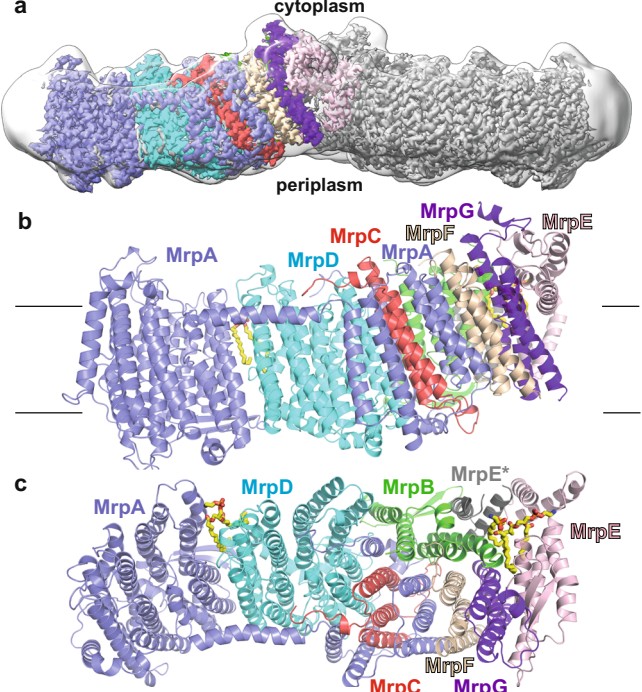

**Fig. 1 | Overall structure of *B. pseudofirmus* Mrp Na⁺/H⁺ antiporter. a** CryoEM density for a dimer, **b** side view with approximate positions of membrane boundaries and **c** top view of a protomer comprising the seven subunits MrpA – MrpG and a partial model of MrpE from an adjacent monomer (MrpE*). In (**a**) the unstructured detergent belt gives an indication of the approximate position of the membrane bilayer.

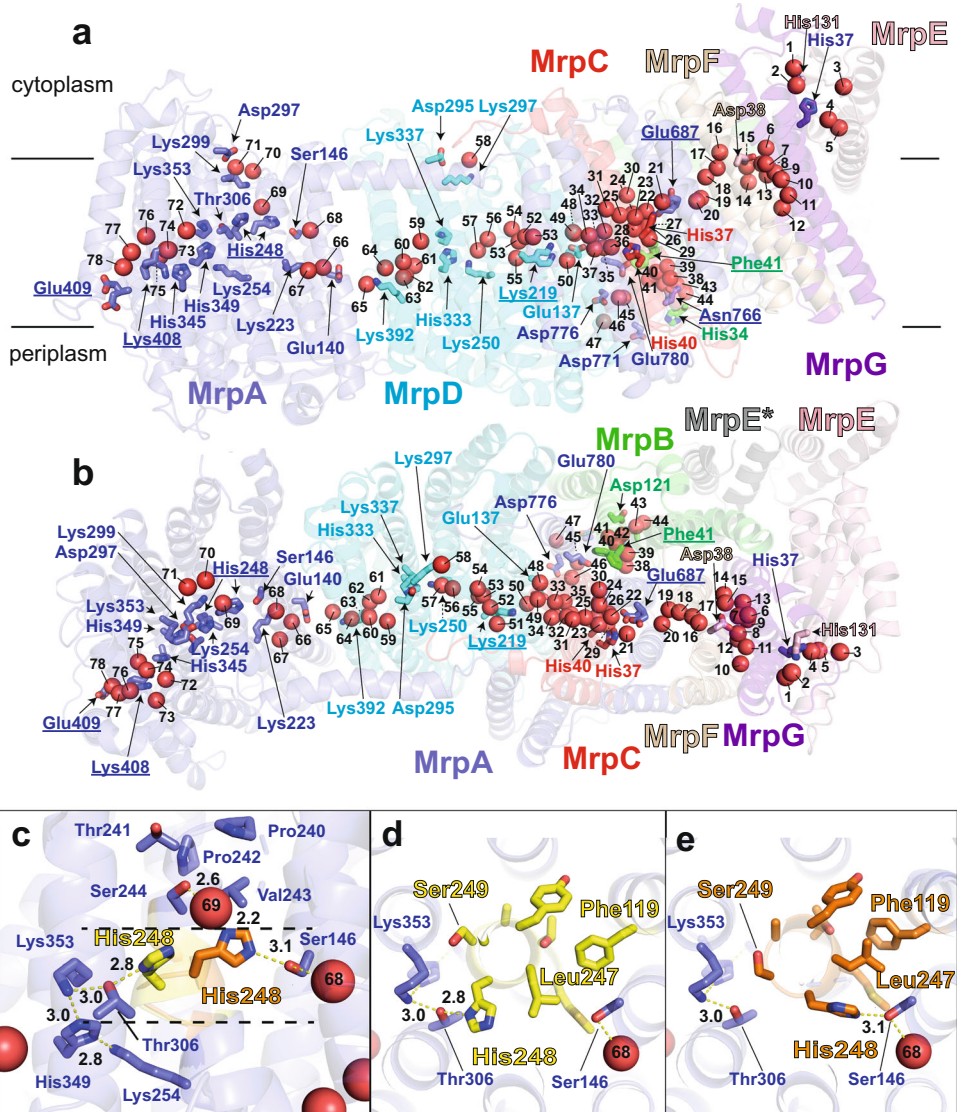

**Fig. 2 | Water molecules, alternative conformations, and functionally important residues. a** Side view and **b** top view of antiporter complex (colors compare Fig. 1) with 78 protein-bound water molecules (red spheres) in putative ion translocation pathways. Selected residues are shown in stick representation, and labels for residues modeled with alternative conformations are underlined. **c** Side view of the protein structure surrounding His248^MrpA. In the *A* conformation (yellow),

His248^MrpA interacts with Thr306^MrpA, in the *B* conformation (orange), it engages in a hydrogen bond with Ser146^MrpA (see Supplementary Movie 1). **d, e** Top view at the level of the mobile segment of TMH8 showing *A* and *B* conformation (colors as in c). The movement of Leu247 is associated with a shift in the side chain position of Phe119^MrpA in TMH4.

## Alternative conformations indicate dynamic regions in the structure

During model building, we noticed clear alternative side chain positions *A* and *B* (called alternate locations or *altloc* in the pdb) for several residues and conformational variations for short sequence stretches (Fig. 2, Supplementary Fig. 2, Supplementary Table 5). In MrpA, the adjacent residues Glu409^MrpA and Lys408^MrpA in TMH12 at the putative periplasmic proton entry site show alternative sidechain positions (Supplementary Fig. 6). However, note that for these two residues, there is a considerable preference for only one of the two conformations (Supplementary Table 5). Corresponding residues, based on MD simulations of bacterial complex I, have been suggested to undergo conformation-dependent proton transfer reactions[27]. Furthermore, we observed a pronounced structural rearrangement in a highly conserved sequence stretch close to the N-terminal end of TMH8 (positions 246 to 252), resulting in alternative positions of several residues, including strictly conserved His248^MrpA (Fig. 2c–e; Supplementary Fig. 2, Supplementary Movie 1). In position *A*, the side chain of His248 points towards

Thr306^MrpA (Fig. 2c, d). In position *B*, it is oriented towards Ser146^MrpA. This group of three residues (His248, Thr306, and Ser146) is strictly conserved in MrpA, and complex I subunit ND5 (Supplementary Fig. 3) and the homologous histidine has been shown to undergo protonation-state dependent conformational changes in complex I MD simulations[27]. Note that also the N-terminal sequence of TMH8 in MrpA (position 240 to 244) is highly conserved, and its two proline residues (position 240 and 242) may form a rigid structure that provides a counter bearing for conformational changes in the adjacent flexible segment (Fig. 2c). In addition, the movement of Leu247^MrpA in TMH8 is associated with a rearrangement of Phe119^MrpA in TMH4 (Fig. 2d).

In MrpD, we modelled the side chain of residue Lys219 in two alternative positions (Fig. 2, Supplementary Figs. 2, 6). This residue is part of the strictly conserved Lys/Glu pair of this subunit that is discussed to play a mechanistic key role in Mrp antiporters and complex I (reviewed in ref. 8).

In the C-terminal domain of MrpA, we modelled Glu687^MrpA in TMH19 with two alternative positions of its side chain (Fig. 2,

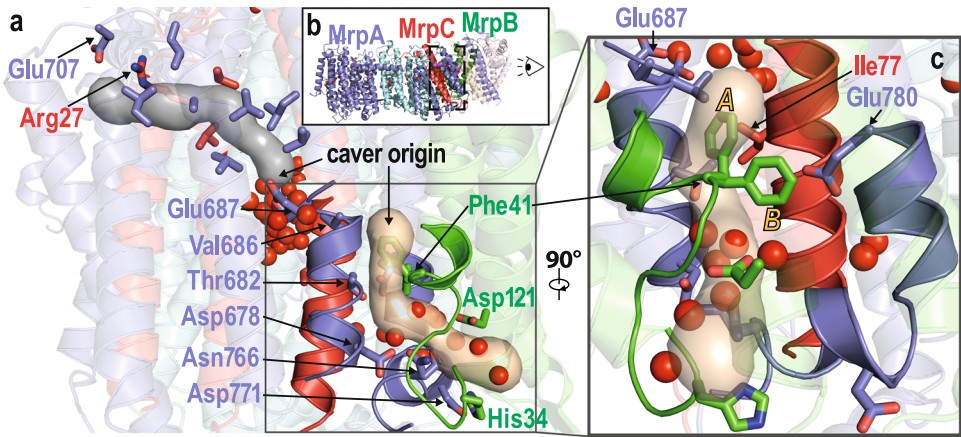

**Fig. 3 | Putative sodium transfer pathways. a** Slice view of the C-terminal domain of MrpA as indicated in (**b**). Tunnels identified with a minimal probe radius of 1.2 Å originating at Glu687[MrpA] (gray) and above Phe41[MrpB] (wheat) are shown in surface view at 20% transparency. The gray tunnel starts at the central water axis around W18 to W31 and terminates at the cytoplasmic side close to Glu707[MrpA] and Arg27[MrpC]. The tunnel is lined by several hydrophobic and poorly conserved residues (Ile650[MrpA], Ala656[MrpA], Val657[MrpA], Val660[MrpA], Val691[MrpA], Leu694[MrpA], Leu704[MrpA], Leu26[MrpC], and Val30[MrpC]). The wheat-colored tunnel originates above Phe41[MrpB] at the level of Glu687[MrpA] but is separated from this residue by the hydrophobic residues Val686[MrpA] and Ile77[MrpC]. The tunnel is lined by the polar residues Thr682[MrpA], Asp678[MrpA], Asn766[MrpA] as well as water W38 to W44 and exits to the periplasm close to His34[MrpC] and Asp121[MrpC]. **c** Detailed view of the putative periplasmic sodium exit channel as seen from MrpB. Strictly conserved Glu780[MrpA] is in close proximity to the tunnel and Phe41[MrpB]; in the *A* conformation, Phe41[MrpB] blocks the tunnel. Highly conserved regions surrounding the putative sodium exit tunnel (MrpA 676-688, MrpA 763-780, MrpB 33-45, and MrpC 66-82, compare Supplementary Fig. 3) are highlighted.

Supplementary Fig. 2). Interestingly, also densities for hydrophobic residues, including Phe41[MrpB] of MrpB showed strong indications for alternative conformations (Fig. 2, Supplementary Fig. 2). Phe41[MrpB] is part of a strictly conserved Pro-Gly-Gly-Gly-Phe sequence stretch in MrpB (Supplementary Fig. 3). Mutagenesis data indicate an important function for Glu687[MrpA] and Phe41[MrpB][15,23,25] and the conformational variations of both residues may be related to sodium translocation (see below).

### Water molecules, buried charges, and polar residues identify ion translocation pathways

Most of the 360 modelled water molecules are located on the hydrophilic protein surfaces exposed to the cytoplasm or periplasm as expected, but about 70 water molecules are found in the interior of the trans-membrane region (Fig. 2, Supplementary Fig. 6). The internal waters are arranged along a central hydrophilic axis that extends over the entire length of the antiporter complex. Our detailed analysis (supplementary text) shows that protonatable and polar residues are equally important for coordinating water molecules in the complex interior, and thus for the formation of ion translocation pathways.

In MrpA, His248[MrpA] is located at the junction of three hydrated pathways. In the *A* conformation, there is a connection via Thr306[MrpA] with a remarkable arrangement of the three strictly conserved residues His349[MrpA], Lys254[MrpA], and Lys353[MrpA] (Fig. 2). From there, water molecules W74 – W78 form a connection to residues Lys408[MrpA]/Glu409[MrpA] at the presumed proton entry site on the periplasmic side. In the *B* conformation, there is a connection via Ser146[MrpA] to a water chain (W66 – W68) that leads to the strictly conserved Glu140[MrpA]/Lys223[MrpA] pair of MrpA. A connection to the cytosolic side is formed by polar residues, including strictly conserved Ser244[MrpA], and associated water molecules W69 – W71 (Fig. 2).

In MrpD, Lys392[MrpD] in TMH12 is in the same position as Lys408[MrpA] in MrpA and the arrangement of the three residues Lys250[MrpD], His333[MrpD], and Lys337[MrpD] corresponds to the Lys-His-Lys triad in MrpA described above (for conservation see supplemental text and Supplementary Fig. 7). As in MrpA, there is a connection of water molecules (W59 – W64) between the TMH12 lysine and the Lys-His-Lys triad in MrpD. In contrast to MrpA, there is no obvious connection by water molecules to the periplasm and there is no residue corresponding to the mobile His248[MrpA] in TMH8 of MrpD. A pathway from the central Lys250[MrpD] residue to the cytoplasm is expected since a corresponding connection has been found in the related complex I subunits ND2 and ND4[21,22]. Nevertheless, there are no water molecules in this region of the structure that would indicate such connectivity. This might indicate that the pathway to the cytoplasm is blocked by the highly conserved Phe341[MrpD] in TMH11 (Supplementary Fig. 3), as recently described for complex I[22]. An overlay of ND2, ND4, and MrpD shows that the position of this residue agrees well with the notion of a closed proton transfer path in the ND4 subunit (Supplementary Fig. 4).

TMH2 and TMH3 of MrpC are located at the center of a highly hydrated region. Waters W32 to W37 sit between MrpC and the strictly conserved Glu137[MrpD]/Lys219[MrpD] pair of MrpD; on the other side, W21 – W31 connect MrpC with the strictly conserved Glu687[MrpA] in TMH19 of the C-terminal domain of MrpA. From here, a string of water molecules W16 – W20 continues to a hydrated region close to Asp38[MrpF]. In agreement with a previous suggestion[14], the pronounced hydration in the vicinity of MrpC suggests a role in sodium transfer, but the water molecules found in the structure did not provide clear information about the connection pathways to the periplasmic or cytosolic side. We have therefore investigated this question further and conducted an analysis of the structure with Caver[28] and MD simulation approaches.

### Cavities in the structure suggest sodium transport pathways

We used the software tool Caver[28] to identify potential sodium transfer pathways with connections to the periplasmic or cytoplasmic side (Fig. 3). Since residues corresponding to Glu687[MrpA] in TMH19 are potentially critical for sodium translocation in Mrp antiporters[14,29] and in MBH[20], we used this residue as starting point for a search with a minimal probe radius of 1.2 Å. We found a connection to the cytosolic side close to Arg27[MrpC] and Glu707[MrpA], but the interior of this cavity at the interface of MrpC and the C-terminal domain of MrpA is predominantly lined by hydrophobic residues (Fig. 3a).

A more extensive system of cavities was found when the probe radius was reduced to 0.8 Å. This revealed a hydrophilic tunnel containing water molecules W16 – 20 leading from Glu687[MrpA] to Asp38[MrpF]. However, access for sodium to Asp38[MrpF] from the cytoplasmic side is not obvious in the Caver analysis.

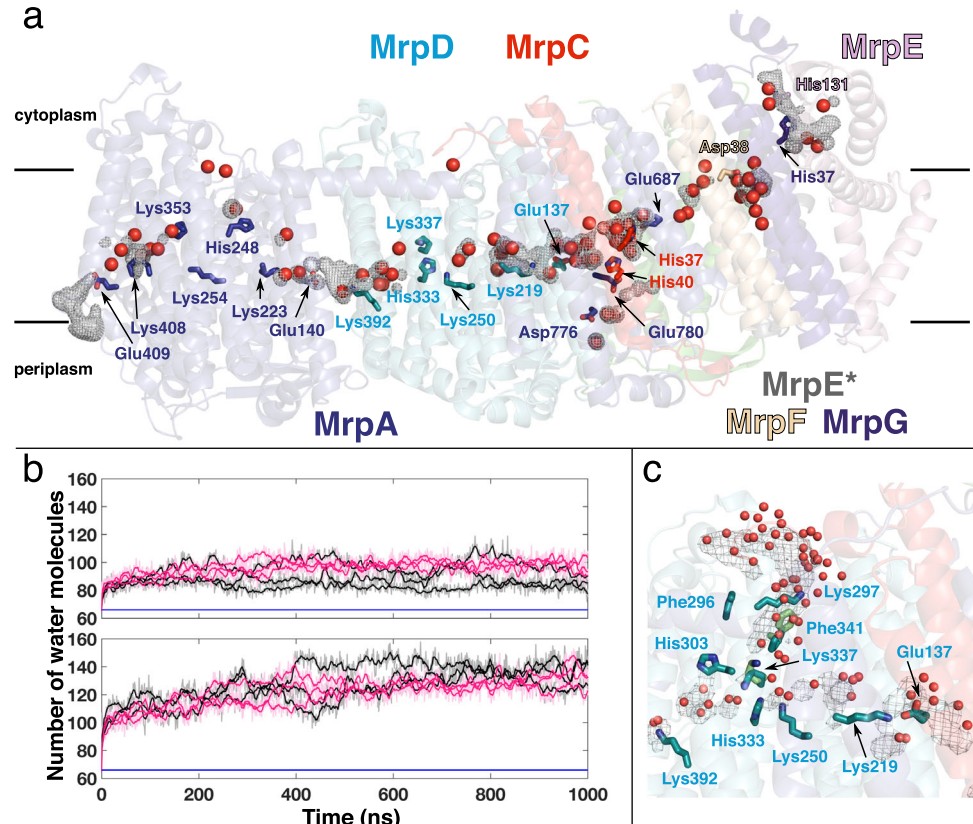

**Fig. 4 | Hydration of Mrp antiporter complex. a** Overall internal hydration in simulation state P, where the protonation state of amino acids is based on pKa calculation, is shown as water occupancy map calculated from all PB1 simulation replicas (dark grey mesh at 20 % isovalue). Internal structural waters (compare Fig. 2) are shown as red spheres. Water occupancy was calculated by selecting waters within 6 Å of residues displayed. **b** Number of internal water molecules in P (top panel) and S (bottom panel) simulation states. Only waters within 6 Å of residues displayed in (a) are counted. Black/pink traces refer to simulations started from A/B alternative conformations observed in the cryoEM structure. Blue horizontal line refers to structural water content (based on the water selection criterion above). **c** Hydration of MrpD subunit in simulation state S, where all amino acids are in their charged state. Transient water-protein-based connectivity forms from the cytoplasmic side to the middle of the protein. The hydration of the region occurs with the movement of Phe341MrpD and Lys337MrpD (cyan – structure, green – simulation). Water occupancy (calculated within 6 Å of residues displayed) is shown as a grey mesh at 20 % isovalue. Water molecules in red spheres are based on a simulation snapshot.

The two water-filled cavities at the MrpA/MrpC/MrpD interface open a putative pathway for sodium from Glu687MrpA to the Lys/Glu pair of MrpD (Fig. 2, Supplementary Fig. 6). For the Mrp antiporter of *A. flavithermus*[14] a connection from the MrpD glutamate to the periplasm was proposed along the interface of MrpA(TMH21)/MrpC(TMH3)/MrpD(TMH5). In the *B. pseudofirmus* structure, Caver does not detect this pathway with reasonable probe radii, as it is blocked by hydrophobic sidechains.

With a probe radius of 1.2 Å we identified a putative sodium-exit tunnel originating above the strictly conserved Phe41MrpB that is separated from Glu687MrpA by the hydrophobic Ile77MrpC and Val686MrpA. This tunnel passes close to Glu780MrpA and is lined by Asp678MrpA, Thr682MrpA, and Asn766MrpA. In the area of the tunnel exit are the residues His34MrpB, Asp121MrpB and Asp771MrpA, and water molecules W38 – W44 (Figs. 2 and 3). In agreement with a proposed function in sodium transfer all aforementioned residues are highly or strictly conserved in Mrp antiporters (Supplementary Fig. 3, Supplementary Table 4). Strikingly, the tunnel is blocked by one of the two alternative conformations of Phe41MrpB (Fig. 3C, Supplementary Fig. 2), suggesting a gating function for this residue, as discussed below.

**Atomistic molecular dynamics simulations of the Mrp complex**
In order to obtain further dynamic insights, we performed fully atomistic MD simulations of the Mrp antiporter in a lipid bilayer in several different states (see computational methods). In the S state, all amino acids were kept in their standard charged states, whereas in the P state, side chain protonation was determined by pKa calculations prior to MD simulation runs (Supplementary Tables 6, 7). The model systems, which are based on high-resolution structural data, show high stability during MD simulations, with root mean square deviation (RMSD) plateauing in the 2-3 Å range (Supplementary Fig. 8b, c). Interestingly, in MD simulations, we observed a reduction in the thickness of the lipid bilayer, especially in the dimerization domain (Supplementary Fig. 8e). This protein-induced variation in membrane width is in good agreement with the detergent belt seen in cryo-EM data (Fig. 1a).

**Hydration of the Mrp complex**
Our MD simulation data show good agreement between the structural water content and the hydration observed in simulation trajectories. In both the S and P states, we observed an increase in the number of internal water molecules in comparison to the structural water content (Fig. 4b). However, the number of internal water molecules plateaus in both states within few hundred nanoseconds, indicating that a stable hydration is reached on these simulation time scales, even when MD simulations were initiated from a totally dry state of the protein (Supplementary Fig. 9e). This suggests that pKa calculations and subsequent MD simulations closely capture the charge and hydration state of the protein. Overall, the hydrated regions overlap very well with structural hydration, especially in the P state simulations (Fig. 4a), and also at a higher threshold (50%) of the water occupancy map

(Supplementary Fig. 9, panels a-d). Even though, some subtle differences exist between the structural water positions and the MD-based water occupancy map, these can be attributed to shorter time scales of MD simulations relative to experiments and also to the fixed point-charge force field used in classical MD simulations (see also[30]). We further analyzed the hydrogen bonding patterns, residence times, and radial distribution functions of water molecules around selected titratable residues that undergo protonation state changes (see Supplementary Note, Supplementary Fig. 10, Supplementary Table 8). These data reveal that depending on the position of amino acids (buried or exposed) as well as their charged state, the behavior of surrounding water molecules can differ (see Supplementary Note).

MrpA and MrpD, which are homologous to complex I antiporter-like subunits (ND2/4/5), have both been suggested to perform proton transfer reactions[14]. Analysis of MD simulation data shows a hydrated connection forms towards the periplasmic side of the membrane in the MrpA subunit with high similarity to a hydrated path observed in complex I subunit ND5 (Fig. 4)[22,27]. Interestingly, in both ND5 and MrpA, titratable residues are conserved (Supplementary Fig. 3) in this domain, and also the distortion of the planar lipid membrane is seen at the proton uptake site (exit route in complex I[22]) (Supplementary Fig. 8d). In contrast to this, a similar connection to the periplasmic side is not observed in MrpD, in part due to the blockage created by hydrophobic residues (e.g. Phe393$^{MrpD}$). This is in agreement with recent high-resolution complex I structures[21,22]. However, at this stage, it cannot be excluded with certainty that a periplasmic proton transfer route opens in the MrpD subunit with some large-scale conformational changes.

A potential connectivity towards the cytoplasmic side of the membrane is seen in the fourteen-helix bundle of MrpA in both the high-resolution cryoEM structure of Mrp complex and its MD simulations (Figs. 2, 4). However, a similar pathway is not observed in the MrpD subunit due to the local sequence and structural variations[27] and also in part due to the blockage created by highly conserved Phe341$^{MrpD}$ residue from TMH11 (Supplementary Figs. 3, 4). The corresponding residue in antiporter-like complex I subunits has been suggested to be a potential gating residue[22]. In our S state MD simulation, where all amino acids are modeled in their standard states, we find transient fluctuations in Phe341$^{MrpD}$ and Lys337$^{MrpD}$ sidechains, which led to water influx in this region and a connectivity between central Lys250$^{MrpD}$ and the cytoplasmic side of the membrane appears (Fig. 4c). The hydration coupled to a conformational change in the region as well as the high conservation of Phe341$^{MrpD}$ (Supplementary Fig. 3) is consistent with its proposed function in gating proton transfer in both complex I and Mrp antiporter families.

### Protonation state-dependent conformational dynamics of His248$^{MrpA}$

Our MD simulations in three different protonation states of His248$^{MrpA}$ allows us to identify atomistic determinants of alternative *A/B* conformations observed in the cryoEM structure. We find that His248$^{MrpA}$ in PA1 and PB1, where this residue is in a charge-neutral state with δ-nitrogen protonated, adopts conformations closer to the structural conformation *A* (Fig. 5, Supplementary Movie 1), with transient hydrogen-bonding interactions to Thr306$^{MrpA}$. However, the *B* conformation in which His248$^{MrpA}$ approaches Ser146$^{MrpA}$ is never observed in these simulations. Therefore, we designed a new set of simulations in which His248$^{MrpA}$ was modeled with its ε nitrogen protonated and a total charge of zero (state PBE in Supplementary Table 6). Remarkably, as revealed by these simulations, His248$^{MrpA}$ now remained stable in the *B* conformation, with a transient hydrogen bond to the side chain of Ser146$^{MrpA}$. In contrast to the predominant *A* or *B* conformations observed in the neutral state of histidine, simulations of doubly protonated His248$^{MrpA}$ (state PBP in Supplementary Table 6) show that it explores intermediate states between the *A* and *B* conformations

(Fig. 5). As observed above for internal protein hydration (Fig. 4), our P-state simulations closely capture the His248$^{MrpA}$ conformational dynamics in contrast to the S-state (Supplementary Fig. 11a).

The protonation-dependent conformational dynamics of His248$^{MrpA}$ allows it to establish a dynamic connection to the three hydrated routes described above, forming a Y-shaped junction (Fig. 5). To probe its potential role in enzyme function, we prepared the His248Ala$^{MrpA}$ mutant and found that the mutated antiporter variant was unable to sustain salt-tolerant growth of *E. coli* strain KNabc[31] that has no intrinsic sodium proton antiport activity (Supplementary Fig. 11b). Additional MD simulations of the His248Ala$^{MrpA}$ mutant showed that in the absence of the histidine side chain, the cytoplasmic connectivity involving a structurally-conserved water is lost (Fig. 5d). We suggest that the loss of His248$^{MrpA}$ and the associated water molecule in the His248Ala$^{MrpA}$ mutant disturbs the proton transfer reactions in this subunit and causes the loss of transport function.

### Putative sodium uptake routes

Our MD simulations show that the side chain of the strictly conserved Glu687$^{MrpA}$ is mobile, and connects to three hydrated cavities; towards the Lys/Glu pair of MrpD, towards Asp38$^{MrpF}$, and also to the cytoplasmic side (Fig. 6a). The latter cavity is consistent with our Caver analysis (Fig. 3), is primarily formed by hydrophobic residues, and hydrates rapidly when Glu687$^{MrpA}$ is modeled as deprotonated (Fig. 6a). The hydrated connection forms between titratable residues at the cytoplasmic side and the Glu687$^{MrpA}$ region (see also Fig. 3 and ref. 20). The extensive conformational dynamics of Glu687$^{MrpA}$ and its accessibility to three hydrated routes suggest that this residue may catalyze the transfer of protons and/or sodium ions. When a sodium ion is modelled at the top of the cytoplasmic cavity near a network of charged residues, it reaches anionic Glu687$^{MrpA}$ rapidly (within ns) and binds there (Fig. 6a). We also modeled a sodium ion near the highly conserved Asp38$^{MrpF}$ and observed that it binds there in a stable manner as long as the residue is in its deprotonated state (Fig. 6b). Upon charge neutralization, Asp38$^{MrpF}$ donates sodium to deprotonated Glu687$^{MrpA}$ (Fig. 6c). Even though a spontaneous sodium uptake from the cytoplasmic side of the membrane was not observed, our MD simulation data suggests that there are two potential cytoplasmic sodium uptake routes (Fig. 6), and both end at the strictly conserved Glu687$^{MrpA}$.

### Gated sodium exit route and its protonation state dependency

Several earlier mutagenesis studies[29,32] have proposed that acidic residues are critical for potential sodium binding sites in the Mrp antiporter complex. Here, our MD simulation data reveal rapid binding (within ns) of sodium ions to residues Glu780$^{MrpA}$ and Asp121$^{MrpB}$ (Fig. 6d), which are both highly conserved and have been suggested to be a part of the possible sodium exit route towards the periplasm. Additional residues that participate in this putative sodium exit route are Asp776$^{MrpA}$, Asp678$^{MrpA}$, Asp771$^{MrpA}$ and His34$^{MrpB}$, and Glu113$^{MrpB}$, where the latter acidic residue is positioned on a β-hairpin scaffold (Fig. 6d). Our MD simulation data are also consistent with our Caver analysis and the presence of water molecules in the structure (Figs. 2 and 3, Supplementary Fig. 6). We observed a clear effect of the charged state of residues Glu780$^{MrpA}$ and Asp121$^{MrpB}$ on the binding of sodium ions. In the case when these residues are modeled charged neutral (P state runs), no sodium binding is observed, revealing that it depends on the protonation state. The binding of a sodium ion to Glu780$^{MrpA}$ is found to be strongly associated with the conformation of strictly conserved Phe41$^{MrpB}$, which has been observed in two alternative conformations in the cryoEM structure (*A* and *B*) (Supplementary Fig. 2). We note that the binding of sodium to anionic Glu780$^{MrpA}$ results in Phe41$^{MrpB}$ turning into the alternative *A* conformation almost immediately regardless of the starting conformation as shown in Supplementary Fig. 12. However, in P state runs, where Glu780$^{MrpA}$ is

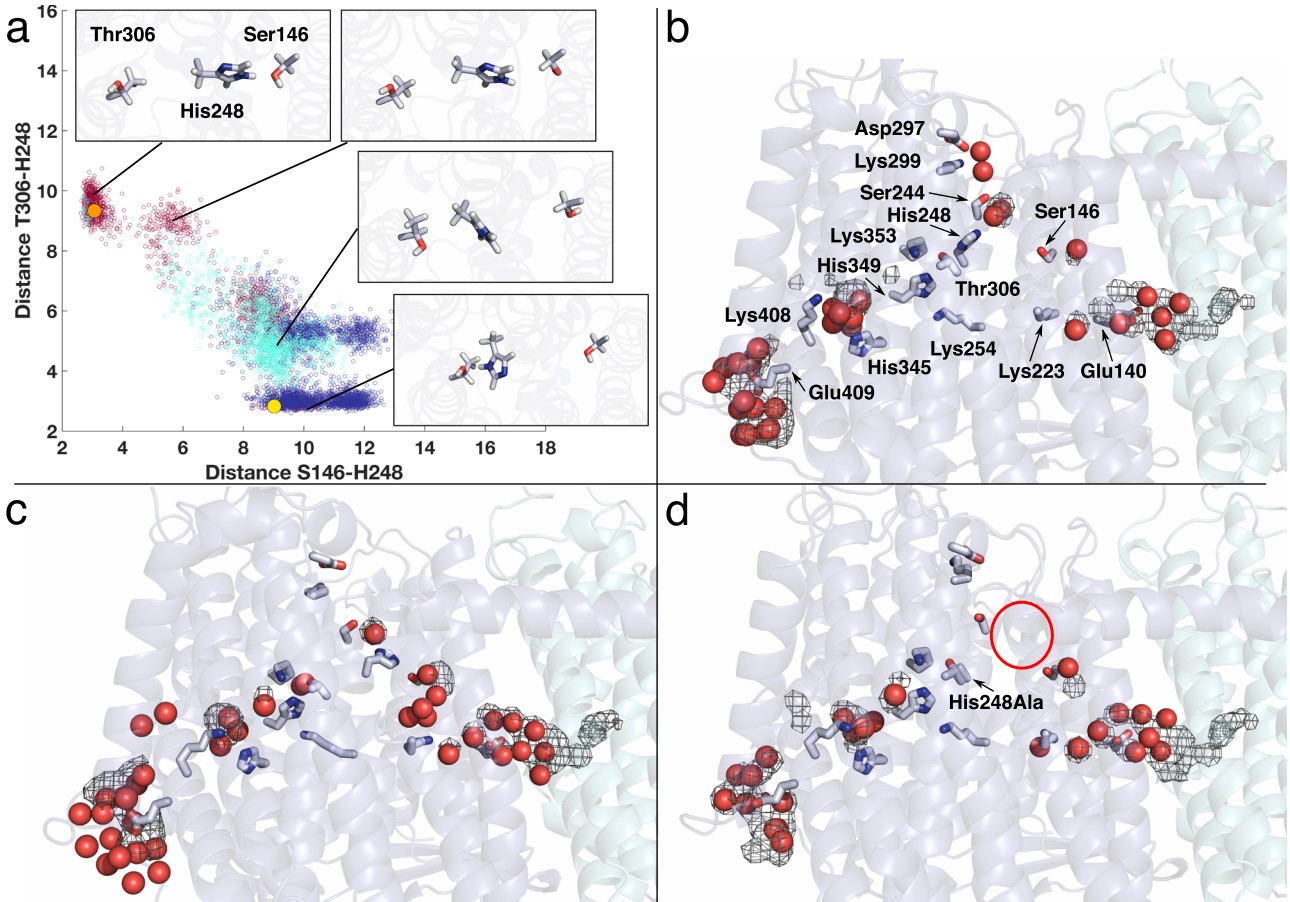

**Fig. 5 | Protonation-dependent conformational dynamics of His248$^{MrpA}$.**
**a** Scatter plot displaying distances between NE2 atom of His248$^{MrpA}$ and OG atoms of Ser146/Thr306$^{MrpA}$ from PBE (neutral His248 with ε nitrogen protonated - burgundy), PBP (doubly protonated His248 - cyan), and PB1 (neutral His248 with δ nitrogen protonated - blue) simulations. Solid yellow and orange spheres indicate the structural distances seen in alternative *A* and *B* conformations (compare Fig. 2 and Supplementary Movie 1), respectively. Representative snapshots from different regions of the scatter plot demonstrate the conformational variety observed in the simulations. Hydration in MrpA subunit with His248$^{MrpA}$ in the *A* state from simulation PB1 (**b**) and in the *B* state from simulation PBE (**c**). His248$^{MrpA}$ in *A*

conformation next to Thr306$^{MrpA}$ connects to the putative proton entry route from the periplasmic side, whereas when it changes to the *B* conformation near Ser146$^{MrpA}$, it connects to the conserved Lys/Glu pair of the MrpA subunit (Lys223$^{MrpA}$ and Glu140$^{MrpA}$). His248$^{MrpA}$ also maintains a cytoplasmic connectivity via a stable water molecule and strictly conserved residues Ser244$^{MrpA}$ and Lys299$^{MrpA}$. **d** Loss of hydration in the region (including structurally-conserved water molecule) near His248Ala$^{MrpA}$ mutant (highlighted by red circle). The water occupancy map is calculated on the basis of criteria defined above for Fig. 4 A/B and is shown at 20% isovalue. The red spheres represent instantaneous positions of water molecules from a simulation snapshot.

charge neutral, alternative conformation *B* is also seen in which Phe41$^{MrpB}$ resides next to neutral Glu780$^{MrpA}$ with no sodium bound. Given the high conservancy of Phe41$^{MrpB}$, these data imply an important role of this aromatic residue in coupling sodium binding to Glu780$^{MrpA}$.

Our MD simulation data suggests sodium ions can travel via two potential routes from the cytoplasm and bind to the first sodium-loading site at the strictly conserved Glu687$^{MrpA}$ (Fig. 6a). But they do not reveal how sodium can travel from the latter site to the second sodium loading site around Glu780$^{MrpA}$, which sits on the exit route towards the periplasmic side (Fig. 6d). Structural and simulation analyses suggest that there are two potential routes that may allow transfer of sodium ions between the two sodium loading sites. One is a direct path that could form upon conformational changes in the highly conserved hydrophobic region comprising Ile77$^{MrpC}$, Val686$^{MrpA}$, and Phe41$^{MrpB}$. Our Caver analysis shows that Phe41$^{MrpB}$ in the alternative *B* conformation unblocks the cavity (Fig. 3), which may allow ion transfer. Consistent with this, our MD simulations suggest that sodium binding at Glu780$^{MrpA}$ triggers Phe41$^{MrpB}$ in *A* conformation (Supplementary Fig. 12), which would block the cavity. This conformational change may therefore be important for preventing back-streaming of sodium ions. A second potential route to transport sodium between

the two sodium-loading sites is through the conserved acidic residue of the Lys/Glu pair of MrpD (Glu137$^{MrpD}$), which is well-connected to Glu687$^{MrpA}$ (Figs. 2, 4, and 6) and is only about 10 Å from the putative release site on the periplasmic side, Glu780$^{MrpA}$. To track this sodium transport pathway, we performed MD simulations by varying the protonation states of amino acid residues (Glu687$^{MrpA}$ and Asp137$^{MrpD}$, see Supplementary Table 6). We observed that the sodium ion can pass on from Glu687$^{MrpA}$ to Asp137$^{MrpD}$ and eventually to Glu780$^{MrpA}$ in a protonation-dependent manner (Fig. 6e, f). Overall, our data suggest that protonation state changes coupled with conformational transitions as well as cooperative sodium binding play a key role in driving sodium translocation.

## Discussion

Under alkaline conditions, *B. pseudofirmus* is dependent on the activity of its Mrp antiporter that allows to stabilize the internal pH value about two units below that of the external environment[1,3,23]. Accordingly, the transfer of protons inward occurs against a concentration gradient and is possible only because the driving force is provided by the electrical component of the membrane potential. The membrane potential of ~ −180 mV (inside negative) is based on the balance between a proton cycle and a sodium cycle. The stoichiometry of the Mrp type

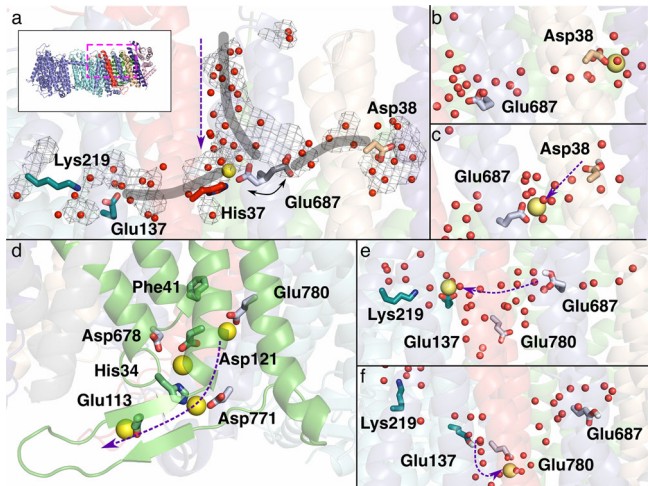

**Fig. 6 | MD simulations of sodium transport in Mrp antiporter. a** The flexibility of Glu687[MrpA] allows it to connect to three hydrated paths (thick grey lines) towards Lys219-Glu137 pair of MrpD, towards Asp38[MrpF], and towards the cytoplasmic side. The vertical purple arrow shows the direction of movement of a possible sodium ion (yellow), bound at Glu687[MrpA], through the hydrated pathway from the cytoplasmic side. **b** Sodium modeled near Asp38[MrpF] binds to its side chain but is transferred to Glu687[MrpA] upon its protonation (**c**). **d** Putative sodium release route with sodium ions bound to conserved residues involved in the path. **e** Protonation of Glu687[MrpA] drives sodium to deprotonated Glu137[MrpD], and upon protonation of Glu137[MrpD], sodium is transported further to Glu780[MrpA] (**f**) through hydrated routes.

antiporters is unknown but is predicted to be electrogenic with the number of transferred protons exceeding the number of transferred sodium ions[1,3,14,15].

We have determined the structure of the Mrp antiporter of *B. pseudofirmus* at 2.2 Å resolution which enabled modelling of 360 water molecules, of which ~70 are located in the trans-membrane region. The large number of internal bound water molecules is unusual for a membrane protein complex and reminiscent of the recent high-resolution structures of respiratory complex I[21,22]. We find that not only titratable residues but also many polar sidechains play a critical role in the extensive internal hydration of the antiporter complex. A substantial number of these polar residues are conserved in Mrp antiporters and respiratory complex I.

Our high-resolution cryoEM map provides evidence for dynamic processes in the transport cycle of the antiporter, some of which are also relevant for respiratory complex I. Of particular interest are the two clearly distinguishable conformations (*A* and *B*) of the MrpA TMH8 segment around His248[MrpA]. By switching between the *A* and *B* conformation, the residue bridges a gap between water-filled proton transfer pathways in the MrpA subunit. Interestingly, His248[MrpA] is at the center of three possible proton transfer pathways. The alternative gating of these pathways is likely to have fundamental significance for the stoichiometry of proton and sodium transport in Mrp (Fig. 7). The preferred conformation of His248[MrpA] depends not only on its protonation state but also on the location of the hydrogen atom on the imidazole moiety. We propose that the ε−nitrogen is protonated when His248[MrpA] is in the *B* conformation. After protonation of the δ−nitrogen via Lys353[MrpA], the proton from the ε−position is delivered to the Lys/Glu pair of MrpA. His248[MrpA] now carries a proton at the δ nitrogen and changes into the *A* conformation. In this state, it receives second proton from the periplasmic side via Lys353[MrpA] and becomes charged (+1e). This drives proton transfer from the Lys/Glu pair of MrpA due to electrostatic repulsion, and the proton on the δ−nitrogen of His248[MrpA] is released to the N side. Neutral His248[MrpA] with its ε−nitrogen protonated reverts back to *B* conformation by hydrogen bonding to

Ser146[MrpA]. The protonation-state dependent conformational dynamics of His248[MrpA] and its connectivity to hydrated and conserved proton transfer pathways are in agreement with earlier MD simulations on bacterial and mitochondrial complex I[22,27]. Furthermore, different complex I structures also indicated two alternative conformations of a corresponding histidine residue (Supplementary Fig. 4). Therefore, we propose that the histidine switch shown in Fig. 7 is a mechanistic feature common to Mrp antiporters and complex I.

We find that Glu687[MrpA] in TMH19 of MrpA has a crucial role for sodium uptake and transfer, in agreement with previous mutagenesis data[25,29]. Remarkably, we observed two alternative conformations for this residue in the high-resolution cryoEM structure, which are likely to be functionally relevant. Extensive side chain fluctuations suggested by our MD simulations are consistent with this observation. An analysis of internal cavities and MD simulation data showed that Glu687[MrpA] is connected to three putative pathways for sodium. The first pathway connects this residue with Asp38[MrpF], a conserved, functionally important residue, as indicated by site-directed mutagenesis[14,15,29]. Steiner and Sazanov proposed an uptake pathway for sodium that starts at the MrpE/MrpG interface and passes Asp38[MrpF] on its way to Glu687[MrpA]. The second path is a more direct connection from the cytoplasm to Glu687[MrpA] through a hydrophobic conduit that has some similarity with a pathway recently described for MBH[20] and the third path leads to the conserved Lys/Glu pair of the MrpD subunit. Our MD simulations suggest that changes in the protonation state of acidic residues (e.g. Glu687[MrpA], Glu780[MrpA] and Asp38[MrpF]) are central for sodium binding and transfer. Moreover, in addition to hydrated paths and sodium binding sites, our combined structural-computational data provide evidence that Phe41[MrpB] might play a central role in a gating mechanism. However, it is not yet clear how sodium traverses the hydrophobic region between the two sodium-loading sites at Glu687[MrpA] and Glu780[MrpA] (SLS1 and SLS2 in Fig. 7). Our Caver analysis shows a tunnel from the periplasm that almost reaches Glu687[MrpA] but is blocked by the hydrophobic sidechains of Ile77[MrpC] and Val686[MrpA]. We speculate that a structural rearrangement might open the hydrophobic blockage, possibly also involving a rotation of TMH19. Note that the corresponding TMH3 of ND6 in complex I has been proposed to undergo a rotational motion during the transition from an α to a π bulge helix[21,33]. MD simulations suggest a possible alternative pathway from Glu687[MrpA] to the periplasm, past the Lys/Glu pair of MrpD. A similar pathway has been suggested by Steiner and Sazanov[14].

Mechanistic schemes for Mrp type sodium/proton antiporters have recently been proposed by Li et al.[15] and Steiner and Sazanov[14]. Several features of the model suggested by Li et al. are clearly at variance with our results. In particular, we consider the proposed uptake of sodium by a pathway in the MrpD subunit to be unlikely because it does not involve Glu687[MrpA] and because it contains numerous positively charged residues. The model proposed by Steiner and Sazanov seems more consistent with our data, and we agree that the Lys/Glu pair of MrpD might play a central role for coupling. However, there are also several important conceptual differences. We propose that protonation or deprotonation of residues in the C-terminal domain of MrpA play a more direct role in coupling. For instance, changes in the protonation state of Glu687[MrpA] are critical for sodium binding and transfer. In fact, a direct involvement of protonation reactions would be more consistent with mechanistic principles operating in other sodium/proton antiporters. However, the exact sequence of events of proton and sodium transfer has to remain speculative at this stage.

In summary, we propose a mechanism that includes the histidine switch as a previously unrecognized device for proton bifurcation in MrpA (Fig. 7). We show that Glu687[MrpA] plays a central role as a loading site for sodium and propose two possible routes for sodium transfer to the cytoplasmic second sodium-loading site centered around Glu780[MrpA]. The protonation state of acidic residues is closely linked to

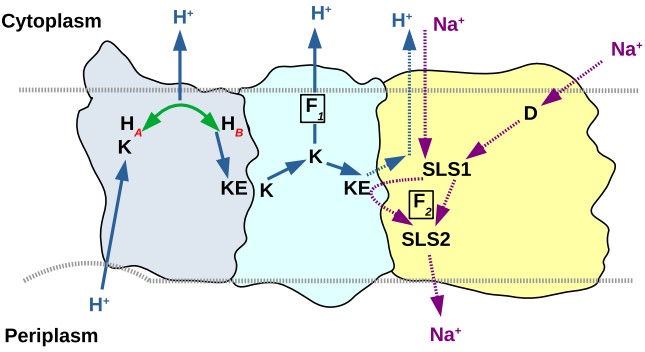

**Fig. 7 | Proposed molecular mechanism of Mrp antiporter.** Sodium ions reach the first sodium-loading site (SLS1) at the strictly conserved and conformationally mobile Glu687$^{MrpA}$ via a hydrated path involving conserved Asp38$^{MrpF}$ (D) or alternatively via a more direct but hydrophobic path observed in the structure and MD simulations (purple dotted vertical arrow). In MrpA (light grey-blue, left), proton transfer occurs from the periplasm to conserved Lys353$^{MrpA}$ (K). His248$^{MrpA}$ accepts the proton from Lys353$^{MrpA}$ and delivers it to the conserved Lys/Glu pair of the MrpA subunit by a histidine switch supported by our cryoEM and MD simulation data (conformational mobility of His248$^{MrpA}$ is shown by a green line and two alternative conformations $H_A$ and $H_B$). Transfer of a second proton to His248$^{MrpA}$ (via Lys353$^{MrpA}$) pushes the proton on the Lys/Glu pair of MrpA to Lys/Glu pair of MrpD (cyan) by electrostatic interactions. The second proton moves energetically downhill from His248$^{MrpA}$ to the cytoplasm. Binding of the first proton to the Lys/Glu pair of MrpD drives the transfer of sodium bound at SLS1 (or at Glu137$^{MrpD}$) to the second sodium-loading site (SLS2), which corresponds to the strictly conserved Glu780$^{MrpA}$, resulting in sodium release on the periplasmic side. The proton moves to the cytoplasm either via MrpD (via the Phe-gated route, marked as $F_1$) or through the hydrophobic pathway (blue dotted vertical arrow). Our model supports electrogenic antiport with a stoichiometry of 2H$^+$ per Na$^+$ driven by the inside negative membrane potential. Importantly, the Lys/Glu pair of MrpD is only ~10 Å from Glu780$^{MrpA}$ (SLS2), which would result in strong electrostatic coupling. Since there are several acidic residues that can bind sodium ions, it is also possible that cooperative sodium binding plays a role in sodium translocation. Proposed proton transfer paths are shown as blue bold arrows (dotted for alternative routes) and sodium translocation paths as magenta arrows. The proposed Phe41$^{MrpB}$ gate is marked as $F_2$.

their function in binding and transfer of sodium, but further work is needed to elucidate the detailed coupling mechanism.

## Methods

### Expression and purification of the *Bacillus pseudofirmus* Mrp antiporter

The *Mrp* operon from *Bacillus pseudofirmus* with its 170 bp preceding sequence was amplified from the genome of *B. pseudofirmus* and subcloned into a pGEM easy vector, with the addition of a His$_6$-FLAG tag at the end of MrpG subunit. We found multiple sequence discrepancies between our cloned gene and the available RefSeq genes, but all of them were either silent nucleotide substitutions or minor changes of non-conserved amino acids. The resulting plasmid was transformed into a modified *E. coli* C41 strain carrying pRARE plasmid, and protein expression was achieved by culturing the transformant in LB medium without any induction for several hours at 37 °C. The bacterial cells were harvested when protein yield saturated, which was around 3-4 hours after the culture reached OD = 0.6. After centrifugation at 5000 g for 12 min, cell pellets were resuspended and lysed by sonication with a probe sonifier for 10 min in lysis buffer [50 mM Tris-HCl (pH 8.0), 150 mM NaCl, and 0.1 mM phenylmethylsulfonyl fluoride (PMSF)] on ice. Cell membranes were pelleted by ultracentrifugation at 125,000 g for 70 min, homogenized in lysis buffer, frozen in liquid nitrogen, and stored at −80 °C.

Protein purification was performed at 4 °C. Membranes were solubilized in 50 mM Tris-HCl (pH 8.0), 150 mM NaCl, and 0.5% lauryl maltose neopentyl glycol (LMNG) for 2 h and insoluble materials were removed by ultracentrifugation at 125,000 g for 32 min. The supernatant was loaded onto chelating agarose resin (Qiagen) coupled with Ni and washed with 5 column volumes 20 mM Tris-HCl (pH 8.0), 150 mM NaCl, 0.05% LMNG, 1 mM beta-mercaptoethanol, and 20 mM imidazole. Bound protein was eluted by increasing the imidazole concentration to 400 mM. The eluate was subjected to size exclusion chromatography on a Superose 6 Increase 3.2/300 column equilibrated in 20 mM Tris-HCl (pH 8.0), 150 mM NaCl, 0.05% LMNG, and β-mercaptoethanol. The fractions containing the Mrp dimer were pooled and concentrated to 1–2 mg/ml for grid preparation.

### Alignments

The Mrp operon from *B. pseudofirmus* was used as input for the BLAST function of UniProt[34]. Each subunit was blasted separately with default conditions targeting the UniRef90 database[35] for 1000 results. Ten additional Mrp operons as well as five Complex I operons (see Supplementary Fig. 3) were further blasted for 100 or 250 results, respectively. The results of each subunit were pooled and edited using the freeware Jalview[36]. Redundant sequences with more than 90% identity and sequences whose length deviated from the mean by more than 10% were removed, resulting in 1000 to 1400 sequences for each subunit. Clustal Omega[37] was used via the JABAWS web-service[38] to perform multiple sequence alignments. The consensus histogram function with logos turned on alongside the preinstalled color scheme ClustalX[39] used with a 50% conservation threshold was utilized for visualization of conserved regions.

### Mutagenesis & growth assay

A plasmid encoding the Mrp antiporter from *B. pseudofirmus* carrying the point mutation H248A in MrpA was prepared by PCR-based site-directed mutagenesis using a pair of primers (TGCATATTTAGCCTC TGCAACAATGGTTAAAGCAGGG and TTGTTGCAGAGGCTAAATATG CACTAACAGGTGTTGGTGC).

*E. coli* KNabc cells[31] were transformed with the plasmid and growth assays were conducted as described by Kosono et al.[32]. Cells were grown for 20 h at 37 °C in LBK-medium supplemented with 0–1000 mM sodium chloride using 2 ml 96-well plates containing, mutant strain, positive and negative controls. The growth was evaluated by measuring OD600 using a SpectraMax M2/M2e Microplate Reader.

### Electron cryo-microscopy (cryoEM)

Quantifoil grids (Cu 400, R 1.2/1.3) were pre-washed in acetone, dried, and glow-discharged twice for 45 sec at 15 mA in a Pelco Glow Discharger. 3 µl of the purified sample were applied onto grids, blotted with a filter paper (Whatman 595) for 3–6 seconds at 6 °C and 100% humidity, and plunged into liquid ethane in a Vitrobot Mark I. For some grids, 1.5 mM fluorinated Fos-Cholin-8 was added to the sample immediately before vitrification, which altered angular distribution of particles and improved data completeness. CryoEM images were acquired from multiple grids using EPU software on a 300 kV Titan Krios microscope operated at 300 kV. Dose-fractionated movies were acquired at 105 k magnification on a K3 camera, resulting in a pixel size of 0.837 Å/pix. The calculated total electron exposure was 50 e$^-$/Å$^2$ on the specimen and the exposure rate was 15 e$^-$/pix/sec on the camera. CryoSPARC Live[40] was used to monitor image quality on the fly.

Single-particle data were processed in RELION 3.1[41], including external programs such as MotionCor2[42] for motion correction and Gctf[43] for contrast transfer function (CTF) estimation. Particles were first picked by a Laplacian-of-Gaussian method implemented in RELION and then by Topaz[44] and subjected to several rounds of 2D and 3D classifications to discard false positive picks. Selected particles were extracted with a box size of 512 px and down-sampled to 360 px, resulting in a pixel size of 1.19 Å/pix. After 3D auto-refinement with a

C2 symmetry, resulting maps and poses were used to correct particle trajectories and higher-order aberrations by using Bayesian polishing and CTF refinement functions in RELION 3.1. During polishing, particles were re-extracted with a box size of 512 px and down-sampled to 400 px, resulting in a pixel size of 1.07 Å/pix, which was used throughout for downstream analyses. The 3D reconstruction of these polished, higher-order aberration-corrected particles provided a consensus map at around 3 Å (Supplementary Fig. 1a). Since this map showed blurry densities at the molecular periphery due to varying angles between the monomers, another two rounds of 3D classification were performed. The resulting more homogeneous class contained 96,337 particles, yielding a dimer map with a relatively even local resolution distribution and a global resolution of 2.96 Å (Supplementary Fig. 1b).

To improve the map quality, we adopted a strategy similar to that used by Steiner et al.[14], whereby particles were symmetry-expanded with a C2 point group and only one monomer was covered for 3D refinement, so that signals from both monomers should align onto the same reference map. This approach improved the 3D reconstruction of the monomer significantly. After correcting for higher-order aberrations again using the pseudo-monomer map and updated particle poses, a round of 3D classification was performed without image alignment to select 513,743 symmetry-expanded particles. The final auto-refinement of these particles with local angular search (0.5 degrees) using SIDESPLITTER[45] yielded a map of the Mrp monomer with an overall resolution of 2.24 Å (Supplementary Figs. 1c, 2a).

### Model building and refinement
An initial homology model was generated in MODELER[46] based on the *Anoxybacillus flavithermus* Mrp monomer (PDB 6Z16) as a template and fitted into our 2.2-Å monomer map in ChimeraX[47]. Iterative model building and refinement were performed in COOT[48] and REFMAC5[49]. The high-resolution map allowed us to fit most of the residues with defined rotamers, as clearly seen for branched-carbon amino acids such as leucine and isoleucine, or for rare backbone conformations of the *cis* prolines. Conversely, some residues showed ambiguous densities, which could be fitted with more than two rotameric states or even with different residue registers (Supplementary Fig. 2b–e). Among these, two alternative main chain traces were possible for residues 246–252 in MrpA, in which His248 alternates its hydrogen-bonding partners. After modelling all the protein residues, the map revealed hundreds of solvent densities to be modelled (Supplementary Fig. 2g). As a guide for modelling solvent molecules, we used Fo – Fc maps calculated in REFMAC5 and servalcat[50], which took into account noise levels of full reconstruction based on the two half-set reconstructions from RELION to calculate a weighted, sharpened map suitable for manual inspection. Water molecules were placed onto Fo–Fc peaks compatible with hydrogen bonding. The number of water molecules were gradually increased over several rounds of model building, refinement, and map inspection until no significant positive peaks were left. Placed water molecules were validated by inspecting Fo–Fc maps after refinement so that none of them showed negative peaks. To identify possible ions (Mg$^{2+}$ and Na$^+$), we inspected the built water molecules according to three criteria: strong density, acidic environment and short bond distances. We first calculated density peak values for each water molecule in ChimeraX and inspected those that have a value greater than 0.04. About 18% water molecules fell into this category, but none of them were in an acidic environment that would favor cation binding. We next inspected residues that are within 2.5 Å from nearby water molecules, possibly representing metal coordination. Forty-two residues fell within this criterion for their main chain or sidechain, but none of them showed a coordination geometry typical for metals. Therefore, we found no strong evidence for bound Mg$^{2+}$ or Na$^+$ ions in our structure. The final atomic model consists of one entire copy of MrpABCDEFG (built as chains A–G), a partial model

of the adjacent MrpE subunit (chain e; only two trans-membrane helices that associated tightly with MrpB were built), three lipid molecules, and 360 water molecules.

### Computational methods
Atomistic molecular dynamics (MD) simulations were performed on the 2.2 Å -resolution structure of Mrp antiporter from *Bacillus pseudofirmus*. The monomeric complex was embedded in a hybrid lipid bilayer that was composed of 80% palmitoyl oleyl phosphatidylethanolamine (POPE) and 20% palmitoyl oleoyl phosphatidylglycerol (POPG). The bilayer was obtained using CHARMM-GUI[51] and protein was inserted into the membrane with the help of OPM alignment[52]. The protein-lipid system was solvated in TIP3 water molecules and 0.15 M NaCl. CHARMM36[53,54] force field was used to obtain the parameters for the protein, lipids, and solvent. NBFIX[55] was applied to treat interactions between sodium ions and acidic residues. As indicated in Supplementary Table 6, in setups S (standard), all amino acid residues were kept in their standard protonation states (e.g. all lysine, arginine, glutamate, and aspartate residues were charged), whereas in setups P, the protonation states of titratable residues were determined by pka calculation[56] on structures in both A and B conformations (see Supplementary Table 7). Supplementray Table 6 also shows the starting conformation chosen for MD simulations, which were performed in both the *A* and *B* conformations.

The model system size was roughly 450,000 atoms, and the simulations were performed with GROMACS 2020.2/2021.5[57]. The model systems were first energy minimized with the heavy atoms of protein constrained by 2000 kJ mol$^{-1}$ nm$^{-2}$ harmonic constraints. Extending the constraints to the phosphorus atoms of the lipids along the membrane normal direction, the temperature was equilibrated to 310 K in NVT ensemble using V-rescale thermostat[58] during a 100 ps simulation. After removing the constraints on the lipid phosphorus atoms, the systems were equilibrated in NPT ensemble using V-rescale thermostat and Berendsen barostat[59] for 1 ns, followed by removal of all constraints and further NPT equilibration for 10 ns. In the production phase, no constraints were applied, and the temperature and pressure were kept at 310 K and 1 atm using the Nose–Hoover thermostat[60,61] and Parrinello–Rahman barostat[62]. The timestep of the simulations was 2 fs which was achieved using the LINCS algorithm[63]. The particle–mesh Ewald method[64] was used to handle electrostatic interactions with a 12 Å cutoff. The van der Waals cutoff was also 12 Å with switching distance at 10 Å. Simulations were analyzed and visualized using software VMD[65] and Pymol[66]. Searches for tunnels and cavities were performed using the plugin CAVER 3.0.3[28] in Pymol[66]. The minimum probe radius was adjusted in a range of 0.7 to 1.2 Å and the maximum distance of the starting point was set to 6 Å, otherwise, default configurations were used.

### Reporting summary
Further information on research design is available in the Nature Research Reporting Summary linked to this article.

## Data availability
The cryo-EM maps generated in this study have been deposited in the electron microscopy data bank (EMDB) under accession code EMD-14124 (monomer) and EMD-15593 (dimer). The atomic model generated in this study has been deposited in the protein data bank (PDB) under accession code 7QRU. The structural data used in this study are available in the PDB under accession codes 7D3U (Mrp antiporter from *Dietzia* sp.), 6Z16 (Mrp antiporter from *Anoxybacillus flavithermus*), 7O71 (respiratory complex I from *Yarrowia lipolytica*), 6I1P (respiratory complex I from *Thermus thermophilus*), 5XTD (respiratory complex I from *Homo sapiens*) and 6ZKA (respiratory complex I from *Ovis aries*). Source data are provided with this paper.

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

## Acknowledgements

This work was supported by the Max Planck Society and grants from the German Research Foundation (Deutsche Forschungsgemeinschaft) (grant ZI 552/5-1 to V.Z.), the Sigrid Jusélius Foundation (senior researcher and project grant to V.S.), the Jane and Aatos Erkko Foundation (project grant to V.S.), the Academy of Finland (grants 294652 and 338176 to V.S.), the University of Helsinki (grant to V.S.), and the Magnus Ehrnrooth Foundation (grant to V.S.) We thank the Electron Microscopy Facility of Max Planck Institute of Biophysics for cryoEM infrastructure and technical support; Sabine Häder and Heidi Betz for help in lab experiments; Juan Castillo, Özkan Yildiz, the Central IT team, and the Max Planck Computing and Data Facility for support in cryoEM data processing. We thank the Center for Scientific Computing (CSC, Finland) for its extensive computational resources. YL was supported by Toyobo Biotechnology Foundation Fellowship and Human Frontier Science Program Long-Term Fellowship.

## Author contributions

Y.L. expressed and purified the protein, prepared cryoEM grids, acquired and processed cryoEM data, analyzed data, drew figures, and contributed to writing the manuscript. O.H. performed MD simulations, analyzed data, drew figures, and contributed to writing the manuscript. A.A. prepared and characterized the mutant, analyzed data, drew figures, and contributed to writing the manuscript. W.K. provided cryoEM infrastructure, supervised the cryoEM work, and contributed to writing the manuscript. V.S. analyzed data and interpreted its mechanistic implications, supervised modeling and simulation work, drew figures, and contributed to writing the manuscript. V.Z. initiated the project, interpreted the mechanistic implications of the structures, drew figures, and contributed to writing the manuscript.

## Funding

## Competing interests

The authors declare no competing interests.
