## [Peer Review File · Nature Communications]

Ion transfer mechanisms in Mrp-type antiporters from high resolution cryoEM and molecular dynamics simulationsREVIEWER COMMENTS

Reviewer #1 (Remarks to the Author):

Lee & Haapanen et al present a high resolution cryo-EM structure together with Caver and MD simulations of a multi-subunit Mrp antiporter complex from *Bacillus pseudofirmus*. A thorough analysis of all of their results revealed details of the antiport mechanism involving proton transfer and sodium translocation. In this team effort, they identified ion translocation pathways as well as key regions of the structure that have mechanistic implications. These noteworthy results add great value to the debated field of ion translocation in complex I and Mrp-type antiporters and will have a significant impact in that field as well as related fields.

Here few suggestions to make this 28-page plus 30+-page SI even better. Overall the language and figures are of high quality with a few exceptions below:

1. The high resolution of 2.2 Å allowed to interpret additional densities outside of the densities for the protein complex as well as multiple conformations of side chains. The authors shared the map and pdb with me and the resolution claim looks reasonable in most regions of the protein but I have a few comments and requests I would like the authors to consider. The data was collected at a pixel size of 0.837 Å/px which would have been great to use instead of that unusual method of downsampling to 1.19 and 1.07 Å/px. For faster processing data can be downsampled or binned and that is typically done to double the pixel size to 1.674 for example. This unusual downsampling unfortunately interferes a little bit with the claimed resolution which is very close to Nyquist. I highly recommend to use the pixel size of 0.837 to avoid getting to close to Nyquist if there is time and resources to do so. For the local resolution in figure S2A, I recommend to include higher as well as lower resolution ranges to be included (e.g. 2.0 – 5.0 Å).
2. It would also be great if the dimer map can be deposited.
3. It would be great if any differences between classes 1,4 and 5 in figure S1B as well as classes 1 and 2 in S1C could be mentioned. Did local resolution filtering reveal any additional information?
4. For Table S1 I recommend to supplement with more information: energy filter yes or no, number of frames, dose rate of 15 e/px/s, number of movies. I also recommend fixing the outliers in the structure and reduce the clash score if possible. Coot has a fantastic way of showing the outliers and residue rotamers can easily be fixed.

5. Coming back to the resolution and the waters. I would tone down the language about how clear the densities for the waters are. They are not and appear in quite high noise level far from a beautiful round shape you would expect. I would not use the words “reliable”, “clearly”. I am glad that the two half maps were looked at as an additional strategy to identify the structural water molecules. I am even more glad to see how simulations nicely complemented the claim of the water molecules, so simply tone down the language about the density for the structured water molecules. I am surprised to see no Mg²⁺ or Na⁺ ions modeled. How sure are the authors all their extra densities are all water molecules? I would inspect the very closely coordinated “water” molecules and check if these could be densities for Na⁺.

6. The membrane region/borders are of interest in this field and I would like the authors to try a little harder to draw the lines for the membranes as accurate as they can based on where they see the densities for the headgroups of the detergent molecule (detergent belt) or their lipids or the data from the simulations where the protein complex was embedded into a lipid bilayer. In Figure 1A a transparent map at a different threshold can be shown and membrane/detergent borders can be indicated. In figure 1B the lines, especially the upper left one under the “cytoplasm” label seems too high. Please check the right side carefully. Please adjust throughout the figures.

7. Check the stereochemistry of important loop region 246-252 of both conformations and fix the bad one. For that particular region it would be great to pull out the two conformations using focused classification if time and resources allow.

8. For all alternative conformations of side chains, please add the occupancy for each. I believe they may exist in the structure but one is clearly more abundant than the 2nd in many cases and remove the ones that are not clear like Phe119A (screenshots attached).

9. Small typos on page 9 (Aps38) 2x, fix to Asp38; page 10 reference to Fig.10? Maybe S10?; “ca. 10A maybe change to ~ or about?, page 13 there are spaces between the number and the % sign 2x, 1 more on page 14, page 14: there is a space between number and degree sign (remove), page 15: wrong reference to S2F, do you mean S2G?;

10. Methods: You did not use Dnase before cell lysis?, no low spin to get rid of cells that were not lysed?, What protein concentration was used for the solubilization with 0.5% LMNG?

11. Why is there “red” in Arg27 in figure 3. That is confusing.

12. Fig. S4 I suggest to add residue labels in G

13. Fig. S6 D: please update with current pdb. That must be a picture using an older pdb since Asn is flipped in the pdb that was shared with me.

14. Figure S7 D, it is hard to see the Q under the green N. Also for ABC, maybe use two different colors for A and C, so the overlap in B looks more clear.

CD2,B/247 LEU/A

CE1,A/119 PHE/A

1.79

3.18

2.88

NZ,B/406 LYS/A

2.94

3.07

2.82

3.19

Reviewer #2 (Remarks to the Author):

The manuscript describes a high-resolution cryo-EM model of Mrp-type antiporters, discusses the placement and protein-coupled dynamics of water and ions, and finally draws inferences in the context of complex I. A major strength of this work is the structural biology of transmembrane or TM systems. Though it is getting more convenient to study TM systems with cryo-EM, the 2.2 Å resolution is definitely a highlight. However, a clear lack of validation of discussed conjectures (a number of which are based on rather limited simulations) really dampens enthusiasm about the work. The concerns are enlisted herewith.

1. "In position A, the sidechain of His248 points towards Thr306MrpA (Fig. 2C, D). In position B, it is oriented towards Ser146MrpA." - Very specific orientations are mentioned. Will be interesting to see the 'resolvability' of the map in these regions using Q-scores [Nature Methods volume 17, pages328–334 (2020)]. This will ensure that the interpretation is physically meaningful and not an indirect manifestation of resolvability differences between positions A and B.
2. How would the authors differentiate between the assignment of waters and ions, also in view of the CAVER results ?
3. The reported 20% occupancy is quite low as an from MD simulations. Unless water molecules are not observed with >50% simulated occupancy, they cannot be assumed as structured water, noting the short ~500 ns time of the simulations. This the experiment-simulation comparisons cannot be made, while the former is depicting somewhat structured water, the later is capturing diffusive water. Recent work on simulation of F or V-type ATPase c-rings can be referred to, where these considerations are discussed.
4. What is the residence time of the water molecules in their positions ? Does the residence change with alteration in the protonation states of the sidechains? A comparison of H-bond patterns between the assigned and simulated water will make the agreement more quantitative, but only after establishing that the residence time of the analyzed water molecules is significant.
5. The statistical significance of the protonation-state dependent His dynamics will have to be thoroughly validated. The results of Fig. 5A might simply be the outcomes of entrapments in local minima corresponding to each protonation state. Enhanced sampling methods such as REST2 (or GaMD) need to be used to indeed confirm the statistical significance of the MD simulations. It is very possible that the distance populations of the conformations can actually mix, but the limited MD time is not adequate for the sidechains to visit conformations that are deviated from the starting point.

6. The work on the ion translocation pathway is even more contentious. It is long known that brute-force MD provides only an incomplete picture of ion translocation (refer to any work by Roux and colleagues). Milestoning simulations using tools like SEEKR are often required to identify such pathways. Otherwise, it is very difficult to comprehend what are the properties of the hydration shell of sodium while they are transported? What is the permeability coefficient? Also, what is the binding affinity of sodium ions to the channel? Do these simulated parameters match with body of existing knowledge on sodium hydration during transport via channels?

7. Also a sequence alignment between complex I and MrpA anti-porters will make the comparisons stronger.

In the absence of the key analyses, I find it very difficult to trust the interpretations of an excellent structural model.

Reviewer #3 (Remarks to the Author):

The work of Lee et al describes the cryo-EM structure and molecular dynamics simulations on the Mrp-type antiporter from *Bacillus pseudofirmus*. The work is very interesting and pertinent.

I have several comments and concerns:

1- The introduction section should include reference to proton and sodium pathways (essential for the transport activity). For a general reader it should indicate what are the structure requirements for a proton and a sodium pathway and what distinguishes them. For a reader specific to the field, proton and sodium pathways proposed in the two other available studies on Mrps should be included. Without this the discussion of ion translocation pathways in the following sections is hard to follow.

2- Sections "Alternative conformations", "Water molecules, buried charges and polar residues identify ion translocation pathways" and "Cavities in the structure suggest sodium transport pathways" could be all integrated in one section "ion translocation pathways" on two sections "proton translocation pathways" and "sodium transport pathways". It would be more consistent and meaningful, considering the function of Mrps.

3- High resolution structure section should be better described: for example, line 95 “anchored by a TMH on the surface of the MrpD subunit” what does surface mean here? Interface? And line 96 “The C-terminal domain of MrpA surrounds MrpC...” what is the composition of the C-terminal domain of MrpA?

4- Section “Water molecules, buried charges and polar residues identify ion translocation pathways”.

a) Are all the pathways described possibly proton or sodium pathways? What distinguishes these?

b) Line 162- “there is no obvious connection by water molecules to the periplasm”. Should it exist? Why? Why not?

c) Line 163 “A connection from the central Lys250 MrpD to the cytoplasmic side is expected” Why is it expected?

d) Line 173 “In agreement with a previous suggestion” from Steiner and Sazanov. The authors should also refer and analyse the previous suggestion from Li et al.

5- The authors used the software tool Caver to identify potential sodium transfer pathways. Was this applied to the hole structure or just to the vicinity of Glu687MrpA? If it was not applied to the hole structure, it should be. If it was, the obtained results should be described. Do the authors exclude other possible sodium pathways? Have they tested the hypothesis from Li et al? The authors suggest Phe41MrpB as a possible gate. Could there be other pathways “blocked” by different gates?

6- At what pH value were the simulations performed?

7- Phe341MrpD is proposed as a gate for proton transfer in MrpD. What is the gate in MrpA?

8- “Protonation state-dependent conformational dynamics of His248MrpA”.

a) Why MD simulations for the other residues with alternative conformations (Lines 115 to 141) are not described? Aren't they relevant?

b) In the mutagenesis study, was protein presence evaluated? How can it be distinguished the presence of a non-functional protein from protein absence in these studies? How can it be distinguished a mutation reflecting stability change versus function?

9- “Putative sodium uptake routes” – Did MD simulations analyse the hole structure? Again, are there other possible sodium pathways?

10- How does the proposed sodium pathways compare with equivalent pathways present in CPA1 and CPA2 members, as NhaA?

11- Discussion section:

a) What is meant by “The alternative gating of these pathways is likely to have fundamental significance for the stoichiometry of proton and sodium transport in Mrp”? What is the stoichiometry proposed?

b) Legend of figure 7. “His248MrpA accepts the proton from Lys353MrpA and delivers it to the conserved Lys/Glu pair of the MrpA subunit...” and “Transfer of a second proton to His248MrpA (via Lys353MrpA) pushes the proton on the Lys/Glu pair of MrpA to Lys/Glu pair of MrpD by electrostatic interactions. The second proton moves energetically downhill from His248MrpA to the cytoplasm.” Why is the first proton from His248MrpA delivered to the Lys/Glu pair of the MrpA and not moved energetically downhill to the cytoplasm as the second proton?

c) I missed a figure/scheme indicated the composition (amino acid side chains and respective distances) of the different proton and sodium pathways.

12- How do the authors think the antiporter activity is activated? How is it regulated? What subunits have these roles? Any clues in the structure for these most biological relevant aspects?

13- What is the reason for the chosen membrane composition (80% POPE and 20% POPG) in the simulations? What is the lipid composition of *Bacillus pseudofirmus*' membranes? And for 150 mM NaCl?

Smaller comments

Mrps are transporters (antiporters) no enzymatic activity is known to be performed.

Line 113 – catalytic cycle, should read transport cycle

Line 259 – enzymatic function, should read transport function

Line 265 – loss of enzymatic activity, should read transport activity

Line 331 – catalytic cycle, should read transport cycle

Line 106 – remove “our”

Line 109 – introduce “sp” in “Dietzia antiporter”

MD calculation – please refer to S and P as systems or states not sometimes systems and other times states.

Reviewer #4 (Remarks to the Author):

This manuscript presents the third structure of an Mrp proton/sodium antiporter, in this case from *Bacillus pseudofirmus*. These antiporters or part of a superfamily of related primary and secondary ion transporters which includes the respiratory Complex I. Hence, information from the "multiple resistance and pH adaptation (Mrp) cation/proton antiporters are of interest to a wide community of researchers. What distinguishes the work presented in this manuscript is that the structure, determined by cryo-electron microscopy is at significantly higher resolution than the other structures of Mrp-type antiporters. The 2.2 Å structure reported here allows the authors to (1) plausibly identify more water molecules internally within the structure; and (2) identify multiple conformational states of amino acid side chains, portions of the polypeptide, and internal water molecules. The additional water molecules include about 70 within putative ion transport pathways. This additional structural detail is tied into MD studies which provide the basis for a model of the sequence of steps in the transporter mechanistic cycle.

The current work is very well done and clearly presented. The additional definition of possible ion transport pathways, including notably the input pathway for Na⁺, add significantly to the literature in this field. The alternative conformations, most noteworthy being the "histidine switch" is also a significant contribution and will be of interest to others.

Several points should be noted by the authors.

1. It is suggested that in Complex I, the idea of four complete and self-contained proton pumping pathways has been questioned by the authors and others, and work on the Mrp complexes will help address this question. The citations to those questioning the four pathways in Complex I do not include the authors. What is the status of this question and does the current work contribute to resolution of this question?
2. The authors make a point distinguishing between electrostatics and the protonation states of critical residues being important to the mechanism of transport. It seems that this is largely a matter of semantics since electrostatics controls protonation states. Please clarify.
3. The authors point out the differences in whether neutral His²⁴⁸(MrpA) is protonated in the delta or epsilon position. If they believe this is important, perhaps a discussion of what factors might shift the preference for the protonation position.

4. The authors suggest that it is a new observation that polar residues, in addition to ionizable residues, are important to stabilize the internal waters. Such a role for polar residues seems obvious, so why is this new?

Overall, this is an excellent paper and worthy of publication in this journal

Response to reviewer comments

Reviewer #1 (Remarks to the Author):

Lee & Haapanen et al present a high resolution cryo-EM structure together with Caver and MD simulations of a multi-subunit Mrp antiporter complex from *Bacillus pseudofirmus*. A thorough analysis of all of their results revealed details of the antiport mechanism involving proton transfer and sodium translocation. In this team effort, they identified ion translocation pathways as well as key regions of the structure that have mechanistic implications. These noteworthy results add great value to the debated field of ion translocation in complex I and Mrp-type antiporters and will have a significant impact in that field as well as related fields.

We thank the reviewer for favorable comments.

Here few suggestions to make this 28-page plus 30+-page SI even better. Overall the language and figures are of high quality with a few exceptions below:

1. The high resolution of 2.2 Å allowed to interpret additional densities outside of the densities for the protein complex as well as multiple conformations of side chains. The authors shared the map and pdb with me and the resolution claim looks reasonable in most regions of the protein but I have a few comments and requests I would like the authors to consider. The data was collected at a pixel size of 0.837 Å/px which would have been great to use instead of that unusual method of downsampling to 1.19 and 1.07 Å/px. For faster processing data can be downsampled or binned and that is typically done to double the pixel size to 1.674 for example. This unusual downsampling unfortunately interferes a little bit with the claimed resolution which is very close to Nyquist. I highly recommend to use the pixel size of 0.837 to avoid getting too close to Nyquist if there is time and resources to do so. For the local resolution in figure S2A, I recommend to include higher as well as lower resolution ranges to be included (e.g. 2.0 – 5.0 Å).

As suggested, we have reprocessed the data at the original pixel size, which was a substantial effort and took nearly two months, but the resolution did not improve beyond 2.25 Å and the map is essentially unchanged. As shown in the figures below, the local resolution maxima remained at 2.19 Å for both cases (1.07 Å/px and 0.837 Å/px), showing that it is the data that is ultimately limiting the resolution and not the downsampling. We also checked the position of water molecules in the new map and found our previous model confirmed. This indicates that the observed water densities were not due to over-interpretation of random noise or map sharpening. Since there are no meaningful differences that would change the model, we used the downsampled map for data deposition and figure presentation. We are glad to share the new map with the reviewer, and will upload it as an associated map to EMDB.

Global FSC curves

Local resolution maps and histograms

2. It would also be great if the dimer map can be deposited.

We are happy to deposit the dimer map as a supplementary/associated map. To avoid repeated file uploads, we would like to do it once the reviewers confirm our refined models.

3. It would be great if any differences between classes 1,4 and 5 in figure S1B as well as classes 1 and 2 in S1C could be mentioned. Did local resolution filtering reveal any additional information?

The classes show differences in angle between the two protomers. This information is now added to the legend of figure S1B. The functional significance of this observation is difficult to judge at this stage and we prefer not to discuss this issue in the main document.

4. For Table S1 I recommend to supplement with more information: energy filter yes or no, number of frames, dose rate of 15 e/px/s, number of movies. I also recommend fixing the outliers in the

structure and reduce the clash score if possible. Coot has a fantastic way of showing the outliers and residue rotamers can easily be fixed.

We have added the requested information in Table S1. We have also fixed rotamers, Ramachandran outliers and clashes as much as possible. The corrected model now shows better scores in common validation metrics.

5. Coming back to the resolution and the waters. I would tone down the language about how clear the densities for the waters are. They are not and appear in quite high noise level far from a beautiful round shape you would expect. I would not use the words “reliable”, “clearly”. I am glad that the two half maps were looked at as an additional strategy to identify the structural water molecules. I am even more glad to see how simulations nicely complemented the claim of the water molecules, so simply tone down the language about the density for the structured water molecules. I am surprised to see no Mg²⁺ or Na⁺ ions modeled. How sure are the authors all their extra densities are all water molecules? I would inspect the very closely coordinated "water" molecules and check if these could be densities for Na⁺.

We agree that there are deviations from the ideal water shape in the map but we have no indications that noise was erroneously modelled as water, as mentioned above for the reprocessed map.

Anyway, we have adapted the description as requested. We note that less round appearance for water molecules may be due to coarse mesh sampling (ChimeraX and Coot use the pixel size as the default mesh sampling). For model inspection, we used maps resampled at finer mesh sizes. To demonstrate this point, we attach the same final map resampled at 0.536 Å/px, which will give a better visual impression of water densities.

When modelling non-protein densities, we remained on the conservative side, meaning that we did not model Mg²⁺ or Na⁺ ions explicitly unless we had strong evidence or prior knowledge to do so. Some studies suggest that the assignment of cations, especially Mg²⁺, is possible by inspecting the cryo-EM maps at different B-factor blurring levels (Wang et al., IUCrJ, 2018), but we found no indication for such strong densities. Na⁺ is more difficult than Mg²⁺ in this regard, since it has only one positive charge. We also have done a systematic analysis of water densities which appear to be stronger than observed for other water molecules and analysed their nearby residues and coordination distances. We again found no closely coordinated densities (with bond distances around 2.4 - 2.5 Å), which might represent Na⁺. We added a more precise description of our criteria for water assignments in Materials and Methods.

6. The membrane region/borders are of interest in this field and I would like the authors to try a little harder to draw the lines for the membranes as accurate as they can based on where they see the densities for the headgroups of the detergent molecule (detergent belt) or their lipids or the data from the simulations where the protein complex was embedded into a lipid bilayer. In Figure 1A a transparent map at a different threshold can be shown and membrane/detergent borders can be indicated. In figure 1B the lines, especially the upper left one under the “cytoplasm” label seems too high. Please check the right side carefully. Please adjust throughout the figures.

We have removed the approximate membrane lines from figures, and instead in Fig. 1A and Fig. S1B we now show the map with the detergent belt to give a better impression of how the protein is embedded into the membrane. The membrane borders are difficult to visualise schematically because MrpA changes the planar geometry as shown in Fig. S8, and MrpE has two unusually short transmembrane helices indicating a thinning of the membrane at the interface of the two protomers. The latter was also observed by Steiner and Sazanov in the dimeric Mrp antiporter of A. flavithermus (ref14).

We now added to the main text on page 3:

“The detergent belt in the dimer map presented in Fig 1A gives an indication of the approximate position and shape of the membrane bilayer and shows thinning of the membrane in the region of the dimerization domain as previously observed for the A. flavithermus Mrp antiporter ¹⁴.”

Fig. S8 shows the curvature of the lipid bilayer caused by the embedded protein structure. In addition, we now analyzed membrane thickness from MD simulation data. In agreement with our previous observations we find that the thickness of the membrane is reduced at the site where protomer contacts would form (see revised Fig. S8).

We added the following passage to the main text on page 7:

“Interestingly, in MD simulations we observed a reduction in the thickness of the lipid bilayer, especially in the dimerization domain (Fig. S8E). This protein-induced variation in membrane width is in good agreement with the detergent belt seen in cryo-EM data (Fig. 1A).”

7. Check the stereochemistry of important loop region 246-252 of both conformations and fix the bad one. For that particular region it would be great to pull out the two conformations using focused classification if time and resources allow.

Following the suggestion, we have fixed the stereochemistry of the region 246-252 of MrpA. The correction includes alleviating a clash between His248 (alt A) and Met237, fixing dihedral angles for Ala250 and fixing non-ideal geometry for Thr252. We note that since the density is a mixture of two conformations, it is difficult to bring the models to perfect stereochemistry while fitting them into the densities.

As for the reviewer’s suggestion to pull out the two conformations by image processing, we had already tried extensive focused classification. We used signal subtraction followed by Class3D with high T value and without alignment (Bai et al., eLife, 2015). However, all efforts ended up in choosing the best classes with two conformations mixed, suggesting that the region is too small to provide enough signal for separation.

8. For all alternative conformations of side chains, please add the occupancy for each. I believe they may exist in the structure but one is clearly more abundant than the 2nd in many cases and remove the ones that are not clear like Phe119A (screenshots attached).

We have now refined the occupancy for each alternative conformation in the new PDB file, using “occupancy refine” and “complete” keywords in refmac, so that the sum of the occupancies for conformations A and B at a given residue range should equal 1.0. The resulting occupancy ratios varied from 0.51:0.49 to 0.83:0.17. Please note that these ratios should not be taken at face value to draw any physical interpretation, because occupancy and B-factor are mutually dependent during refinement and the change in one value can easily be absorbed by the other. The table listing occupancies is added to the supplement (Table S5).

On page 4, we have added the information that for Glu409 and Lys408 in MrpA there is a clear preference for only one of the two conformations.

“In MrpA, the adjacent residues Glu409^{MrpA} and Lys408^{MrpA} in TMH12 at the putative periplasmic proton entry site show alternative side chain positions (Fig. S6). Note however that for these two residues there is a clear preference for only one of the two conformations (Table S5).”

We think that the two conformations for Phe119^{MrpA} are real as shown below. The density deviates from an ideal phenylalanine shape having one defined conformation, and if we model only one conformation, we observe a significant Fo-Fc peak indicating the second location. Furthermore, the

existence of two alternative conformations for Phe119^{MrpA} is supported by the fact that if only one conformation is modeled, steric conflict with one of the alternative conformations of residues Leu247^{MrpA} and Thr251^{MrpA} occurs.

Down-sampled map:

Original map:

9. Small typos on page 9 (Aps38) 2x, fix to Asp38; page 10 reference to Fig.10? Maybe S10?; "ca. 10A maybe change to ~ or about?, page 13 there are spaces between the number and the % sign 2x, 1 more on page 14, page 14: there is a space between number and degree sign (remove), page 15: wrong reference to S2F, do you mean S2G?;

Thank you, we have corrected typos and references.

10. Methods: You did not use Dnase before cell lysis?, no low spin to get rid of cells that were not lysed?, What protein concentration was used for the solubilization with 0.5% LMNG?

We did not use DNase before cell lysis and no low spin was performed. We also did not quantify total protein concentration before solubilization as our routine practice. Instead, a total solubilization volume was adjusted on the basis of the culture volume. This protocol was optimised using fluorescence-detection size-exclusion chromatography of ml-scale cultures. We find that 0.5% LMNG is fairly robust in solubilizing membranes from various amounts of starting materials.

11. Why is there “red” in Arg27 in figure 3. That is confusing.

Arg27 belongs to MrpC which is depicted in red with nitrogens shown in blue.

12. Fig. S4 I suggest to add residue labels in G

We have added residue labels in Fig. S4G.

13. Fig. S6 D: please update with current pdb. That must be a picture using an older pdb since Asn is flipped in the pdb that was shared with me.

In fact, the side chain was flipped. We have corrected the figure.

14. Figure S7 D, it is hard to see the Q under the green N. Also for ABC, maybe use two different colors for A and C, so the overlap in B looks more clear.

We have amended Fig. S7 accordingly.

Reviewer #2 (Remarks to the Author):

The manuscript describes a high-resolution cryo-EM model of Mrp-type antiporters, discusses the placement and protein-coupled dynamics of water and ions, and finally draws inferences in the context of complex I. A major strength of this work is the structural biology of transmembrane or TM systems. Though it is getting more convenient to study TM systems with cryo-EM, the 2.2 Å resolution is definitely a highlight. However, a clear lack of validation of discussed conjectures (a number of which are based on rather limited simulations) really dampens enthusiasm about the work. The concerns are enlisted herewith.

We thank the referee for comments and excellent suggestions. We have now answered below all the concerns raised and amended the manuscript accordingly, with additional simulation and analysis.

1. "In position A, the sidechain of His248 points towards Thr306MrpA (Fig. 2C, D). In position B, it is oriented towards Ser146MrpA." - Very specific orientations are mentioned. Will be interesting to see the 'resolvability' of the map in these regions using Q-scores [Nature Methods volume 17, pages328–334 (2020)]. This will ensure that the interpretation is physically meaningful and not an indirect manifestation of resolvability differences between positions A and B.

We have calculated Q-scores for the MrpA region 248-252, as shown in the table below. The result shows that His248 has side chain Q-scores better (0.7525 and 0.7523 for conformations A and B respectively) than the expected value at 2.24 Å resolution (0.7216), supporting our assignment for their specific orientations. Please note however that we could only separately calculate the scores for each conformation and thus would expect lower values than ideal, since the "resolvability" assumes that the density is a representation of one underlying atomic model, while our density has two underlying models. As far as we know, the Q-score metric has not been applied or tested against the data with alternative conformations, and therefore we would like to not present it in the main manuscript. On a related note, we have performed occupancy refinement as per another reviewer's request and now all residues with alternate conformations are assigned a pair of occupancies (sum equaling 1.0), which would supplement the point raised by the reviewer. Please see Table S5.

	Chain	Res	Res #	Q_backBone	Q_sideChain	Q_residue	ExpectedQ@2.24
alt_A	A	TYR	246	0.819738	0.813364	0.815488	0.7216
	A	LEU	247	0.704323	0.555226	0.629775	0.7216
	A	HIS	248	0.78878	0.75247	0.766993	0.7216
	A	SER	249	0.602808	0.617722	0.607779	0.7216
	A	ALA	250	0.585751	-0.064609	0.455679	0.7216
	A	THR	251	0.699946	0.360253	0.554363	0.7216
	A	MET	252	0.809223	0.685115	0.747169	0.7216
alt_B	A	TYR	246	0.82959	0.846789	0.841056	0.7216
	A	LEU	247	0.68421	0.806027	0.745118	0.7216
	A	HIS	248	0.576303	0.752328	0.681918	0.7216

	A	SER	249	0.805713	0.414895	0.67544	0.7216
	A	ALA	250	0.70043	0.575041	0.675352	0.7216
	A	THR	251	0.738656	0.684351	0.715383	0.7216
	A	MET	252	0.830098	0.761616	0.795857	0.7216

2. How would the authors differentiate between the assignment of waters and ions, also in view of the CAVER results ?

When modelling non-protein densities, we stayed on the conservative side, meaning that we did not model ions explicitly unless we had strong evidence or prior knowledge to do so. We also analyzed water densities systematically. We found that they appear to be stronger than observed for other water molecules and analyzed their nearby residues and coordination distances. We again found no closely coordinated densities (with bond distances around 2.4 - 2.5 Å), which might represent an ion. Thus, we are convinced that our original assignments of water molecules are correct. We added a more precise description of our criteria for water assignments in Materials and Methods.

The Caver software searches for internal volumes in protein structures that have a connection to the external medium. The program does not take into account the chemical nature of the residues lining the cavity and it does not provide information on the content of a cavity. A cavity or tunnel is displayed if at the narrowest point the volume does not fall below a certain probe radius. Of course, this does not directly mean that it must be a transport path for ions. On the other hand, a potential pathway with a smaller minimal diameter cannot be discarded because dynamic changes of the protein structure may still allow opening of a narrow passage. The ionic radius of sodium without its hydration shell is given as 0.95 to 1.05 Å. This correlates to a coordination number of 4 to 6 in crystals (it can go up to 1.39 Å at coordination number 12 according to the CRC Handbook of Chemistry and Physics). With a hydration shell, values from 2.2 to 2.4 Å are reported in the literature [Fifen & Agmon J. Chem. Phys., 2019 [<https://doi.org/10.1063/1.5020150>]]. The pathway to Glu687 in MrpA, which we discuss as a possible sodium entry route, was found with a probe radius of at least 1.1 Å by Caver. The exit-tunnel around MrpB-F41 is around 1.0 Å in all complexes where MrpB-F41 is in the B conformation (this includes all published Mrp structures). In conclusion, the potential pathways for sodium discussed here are wide enough for a sodium ion to enter but additional information is needed to monitor sodium transport.

3. The reported 20% occupancy is quite low as an from MD simulations. Unless water molecules are not observed with >50% simulated occupancy, they cannot be assumed as structured water, noting the short ~500 ns time of the simulations. This the experiment-simulation comparisons cannot be made, while the former is depicting somewhat structured water, the later is capturing diffusive water. Recent work on simulation of F or V-type ATPase c-rings can be referred to, where these considerations are discussed.

We extended all P and S state simulations to 1 microsecond (three simulation replicas each). The new data (Fig. 4 revised) show that protein hydration has already stabilised in < 500 ns, in both P and S states and in all simulation replicates, which is also in agreement with our earlier work on respiratory complex I (Haapanen and Sharma, Sci. Rep. 2017) that waters rapidly hydrate the protein on these timescales.

Second, we recalculated the volumetric map using updated simulation sampling (Fig. 4 revised). The map does not change further with enhanced sampling. These data suggest that as far as hydration of the protein interior is concerned, the simulated time scales are sufficient and reproduce well the structural water content in the protein interior.

We also analysed hydration at 50% occupancy as the referee suggested. Since the fine details of the 50% occupancy map are difficult to visualise in an overall figure of the antiporter complex, we provide a new figure (new Fig. S9) and show that even at 50% occupancy, several structurally conserved water positions are very well reproduced by simulations.

To further consolidate our findings, we also performed two 1-microsecond MD simulations starting with completely dry protein. As above, within hundreds of nanoseconds, protein hydration reaches a plateau, and approaches structural water content in simulation of P states (Fig. S9E).

4. What is the residence time of the water molecules in their positions ? Does the residence change with alteration in the protonation states of the sidechains? A comparison of H-bond patterns between the assigned and simulated water will make the agreement more quantitative, but only after establishing that the residence time of the analyzed water molecules is significant.

We calculated residence times of water molecules, based on hydrogen-bonding dynamics and life times, around several titratable residues that show non-standard protonation states (e.g. neutral state of Lys or Glu). Since we performed MD simulations in both states, we also analysed residence times for both states (neutral and charged). We have added the new Table S8 to summarize the results.

Yes indeed, and as shown in Table S8, depending on the type of the residue and its location, we do see a whole range of residence times, which also change with changes in protonation state.

Our analysis revealed that for residues that face bulk-like surroundings, water molecules have similar residence times (<ns), irrespective of the protonation state modelled or simulated (neutral or charged) (see Table S8, blue color). However, there are residues (see Table S8, green color) where a change in protonation state alters the residence times of water molecules around them.

Interestingly, relatively longer hydrogen-bonding residence times of water molecules were observed around residue Glu780. This residue binds a sodium ion in its simulated charged state, and residence times of water molecules do not differ when its neutral state is simulated, i.e. when it does not bind a sodium ion (Table S8, dark orange color). This is in part due to the a hydrogen atom in its neutral state being replaced by a tightly bound sodium ion in its charged state.

Furthermore, we noticed unusually long hydrogen bonding residence times (>ns) of water molecules around several amino acid residues (Table S8, red color). In several of these cases, a structurally resolved water was stable in the vicinity of the residue, yielding a longer hydrogen bonding residence time.

To further aid in our understanding of water behaviour around residues simulated in two states, we also calculated the radial distribution function of water (around the residue) and hydrogen bond between water and residues. The data in Fig. S10 shows clear differences in hydrogen bonding patterns and radial distribution functions between charged and neutral states simulated.

To make these points clearer we have added the paragraph "Protein hydration and residue-water bonding analysis" to the supplement.

5. The statistical significance of the protonation-state dependent His dynamics will have to be thoroughly validated. The results of Fig. 5A might simply be the outcomes of entrapments in local minima corresponding to each protonation state. Enhanced sampling methods such as REST2 (or GaMD) need to be used to indeed confirm the statistical significance of the MD simulations. It is very possible that the distance populations of the conformations can actually mix, but the limited MD time is not adequate for the sidechains to visit conformations that are deviated from the starting point.

We have now performed GaMD simulations to assess the behaviour of His248 dynamics. Due to the large size of the system and related potential convergence issues, we performed these simulations on a truncated system consisting only of MrpA subunit. We performed simulations in P state to compare to Fig. 5A, but also neutralized the acidic residue facing the membrane to prevent unnatural hydration in the region. We performed three independent simulation replicates (each 400-500 ns long) in four different states (two protonation states of neutral His248 with proton on delta and epsilon nitrogen, and in two alternative conformations A and B). We observed a standard deviation of bias potential around 2 kcal/mol in all cases, hence applying a reweighting procedure to obtain free energies should be fine. The data in the figure below show 2D-potential of mean force of His248 conformational dynamics in states with regard to two distances (as in Fig. 5A). RC1 - His248/Ser146 and RC2 - His248/Thr306 distances. In panels A and B, the His248 is in neutral state with delta nitrogen protonated starting from A and B alternative conformations, respectively. In panels C and D, His248 is in neutral state with epsilon-nitrogen protonated. The comparison of panels A and D shows that although the free energy basins overlap for some regions (which is in partial agreement with unbiased MD data, Fig 5A, where conformations mix showing equilibrated behaviour), there is a clear shift in the energy basin (occupancy) towards shorter His-Ser distance (RC1), when His248 is with epsilon-nitrogen protonated. On the other hand, it is the shorter His-Thr distance (RC2) that is much more likely in the delta-nitrogen protonated case. This is in agreement with correspondence between the position of the proton on the delta- and epsilon-nitrogen and A and B conformations, respectively.

Even though GaMD simulations support the conclusions qualitatively, the overlap between two energy basins and accurate estimate of the barrier between the two states may depend on other factors. For instance, the dynamics of His248 is not only dependent on its own protonation state, but also on that of the surrounding residues (see Fig. S11, updated numbering). Therefore, we would prefer to not include GaMD simulation data in the current manuscript, and instead deal with the problem in a more quantitative and systematic manner, also with umbrella sampling and its variants, in a separate future study.

Please note that we have now extended the unbiased MD simulations to longer time scales (ca. 1 microseconds with three independent simulations replica each). The updated figure 5 shows the same trend as before.

Overall, our unbiased MD and GaMD data suggest that the barrier between the two conformations may not be high, but occupancy differs. Thus, the position of His in A and B conformation is not because of the entrapment in local minimum but is linked to its own protonation state as well as the protonation state of neighboring residues. This is expected of a residue that is most likely involved in gated proton transfer, as proposed here.

6. The work on the ion translocation pathway is even more contentious. It is long known that brute-force MD provides only an incomplete picture of ion translocation (refer to any work by Roux and colleagues). Milestoning simulations using tools like SEEKR are often required to identify such

pathways. Otherwise, it is very difficult to comprehend what are the properties of the hydration shell of sodium while they are transported? What is the permeability coefficient? Also, what is the binding affinity of sodium ions to the channel? Do these simulated parameters match with body of existing knowledge on sodium hydration during transport via channels?

We agree with the reviewer that brute-force MD simulations (or even some advanced simulations methods) are not sufficient to provide a full detailed picture of ion translocation. However, our proposals on sodium transfer paths and binding sites (Fig. 6) are all built on combining the mutagenesis, structural and simulation data available so far. Thus, on one hand we agree that even though a full ion translocation picture is not obtained here, our work nevertheless provides convincing new data on the sodium exit route. For instance, our MD simulations consistently show binding of sodium ions (in protonation state dependent fashion, please see Fig. 6D and text), which has not been observed before. This looks very promising and is also in agreement with earlier biochemical data. This forms the basis to further test our proposal with more advanced experimental and computational methods in the future.

The additional sodium transfer paths we discussed in Fig. 6A-C and E/F, where protonation-state-dependent binding and unbinding of sodium ions has been simulated, are all localized to regions where conserved acidic residues are found in MrpA/F subunits, and which have also been proposed to bind sodium ions in earlier biochemical studies. Our work takes a step forward and shows, by simulating the high-resolution structure in different charged states of the enzyme, that indeed sodium ions can travel by binding/unbinding in these regions. Thus overall, our results of spontaneous binding of sodium ions in a putative exit route (Fig. 6D) and internal sodium transfer routes (Fig. 6A-C and E/F) provide novel atomistic insights into sodium transport by Mrp antiporters. But obviously, further evidence is needed to prove or disprove these, and we will address these open issues in future work.

We emphasize that very little is known about the sodium binding sites and sodium translocation paths in Mrp antiporters, since no sodium ions have been identified in the structures obtained so far. We are also not aware of any data on binding affinity and permeability coefficients of sodium ions in Mrp-type antiporters. However, we further analyzed our simulation data and observed that ions modeled near residues Glu687 or Asp38 (see Fig. 6A-C) interact with fewer water molecules compared to sodium ions in the bulk aqueous phase. This means that there is partial dehydration of sodium ions when they are bound to the protein interior, and are compensated by anionic side chains of acidic residues. This is in agreement with earlier studies on sodium channels, where similar effect is seen, albeit architecture of sodium binding site is completely different in the latter system. See for instance ref. Chen, Brooks and Damjanovic, Biophys J, 120, 15, 3050-3069.

7. Also a sequence alignment between complex I and MrpA anti-porters will make the comparisons stronger.

We show a sequence alignment between complex I and Mrp antiporters in supplemental figure S3. We have made the labelling clearer. Moreover, an extensive analysis of conservation of residues in putative ion translocation pathways in Mrp antiporters and complex I is shown in Table S4.

Reviewer #3 (Remarks to the Author):

The work of Lee et al describes the cryo-EM structure and molecular dynamics simulations on the Mrp-type antiporter from *Bacillus pseudofirmus*. The work is very interesting and pertinent.

We thank reviewer 3 for favorable comments.

I have several comments and concerns:

1- The introduction section should include reference to proton and sodium pathways (essential for the transport activity). For a general reader it should indicate what are the structure requirements for a proton and a sodium pathway and what distinguishes them. For a reader specific to the field, proton and sodium pathways proposed in the two other available studies on Mrps should be included. Without this the discussion of ion translocation pathways in the following sections is hard to follow.

We have expanded the introduction, but we cannot include a comprehensive discussion of sodium and proton transfer pathways due to space limitations. This would be a topic for a stand-alone review.

On page 2, we added information on sodium and proton transfer pathways in the well characterized single-subunit sodium proton antiporter NhaA and specified two previous proposals for Mrp antiporters:

“In the single-subunit antiporter NhaA, a few closely spaced residues in the center of the protein have a key function for binding of sodium ions or protons and the transported ions move along the same funnel-like pathways to either side of the membrane⁴. In contrast, the ion translocation pathways in the multi-subunit protein complexes of the complex I superfamily and in Mrp-type antiporters are debated. For the antiporter, several mutually exclusive models have been suggested^{5,14,15,18}. Steiner and Sazanov propose that transmembrane proton transfer is carried out by the MrpA and MrpD subunits while sodium is taken up from the cytoplasm by MrpE and released to the periplasm at the interface of MrpA, MrpC and MrpD¹⁴. In contrast, Li et al.¹⁵ assign proton translocation to MrpA and sodium translocation to MrpD with an exit pathway similar to that proposed independently in ref¹⁴.”

2- Sections “Alternative conformations”, “Water molecules, buried charges and polar residues identify ion translocation pathways” and “Cavities in the structure suggest sodium transport pathways” could be all integrated in one section “ion translocation pathways” on two sections “proton translocation pathways” and “sodium transport pathways”. It would be more consistent and meaningful, considering the function of Mrps.

Thank you for this suggestion but we prefer to keep our original paragraph division. It is a basic assumption that transport pathways for protons and sodium are separate. For the results section, we prefer a more neutral point of view and draw our conclusions in the discussion.

3- High resolution structure section should be better described: for example, line 95 “anchored by a TMH on the surface of the MrpD subunit” what does surface mean here? Interface? And line 96 “The C-terminal domain of MrpA surrounds MrpC...” what is the composition of the C-terminal domain of MrpA?

We have amended the text accordingly.

4- Section “Water molecules, buried charges and polar residues identify ion translocation pathways”.
a) Are all the pathways described possibly proton or sodium pathways? What distinguishes these?

In our opinion it is not straightforward to conclude this based on structural information alone. However, our MD simulations were helpful here. For instance, we consistently observed spontaneous binding of sodium ions to selected acidic residues in a protonation-state-dependent fashion (Fig. 6D). Given the known importance of some of these residues from earlier biochemical studies, we think our simulations suggest that this is a putative sodium exit route. Note, however, that sodium binding/unbinding may also be driven by the protonation state, thus, some of the routes may be shared. Moreover, we also applied knowledge on respiratory complex I to assign putative proton transfer paths (e.g. MrpA and MrpD, which are homologous to ND5 and ND4/2 of mitochondrial complex I, respectively).

During revision of our manuscript, and also in response to another referee, we performed additional analyses to identify sodium ions in our cryo-EM maps. But there is no convincing evidence to change our assignment of water molecules, given that, we build our arguments based on simulation data, earlier biochemical data as well as comparison to respiratory complex I.

b) Line 162- “there is no obvious connection by water molecules to the periplasm”. Should it exist? Why? Why not?

We refer to the similarity between MrpA and MrpD. We have described the connection to the periplasm in MrpA in detail. Although the core fold of MrpA and MrpD are superimposable and strictly conserved Lys residues are present in strategically important positions, a connection by water molecules is only found in MrpA. We think this is remarkable.

c) Line 163 “A connection from the central Lys250 MrpD to the cytoplasmic side is expected” Why is it expected?

It is expected because of the similarity between MrpD and complex I subunits ND2 and ND4 for which connections from the strictly conserved central Lys to the matrix (corresponding to cytoplasm) have been shown. We have added this information and the text reads:

“A pathway from the central Lys250^{MrpD} residue to the cytoplasm is expected, since a corresponding connection has been found in the related complex I subunits ND2 and ND4^{21,22}. Nevertheless, there are no water molecules in this region of the structure that would indicate such a connectivity.”

d) Line 173 “In agreement with a previous suggestion” from Steiner and Sazanov. The authors should also refer and analyse the previous suggestion from Li et al.

We refer to the work of Li et al. in the introduction and in the discussion. We have now added more detailed information on their model in the introduction (see above).

5- The authors used the software tool Caver to identify potential sodium transfer pathways. Was this applied to the hole structure or just to the vicinity of Glu687MrpA? If it was not applied to the hole structure, it should be. If it was, the obtained results should be described. Do the authors exclude other possible sodium pathways? Have they tested the hypothesis from Li et al? The authors suggest Phe41MrpB as a possible gate. Could there be other pathways “blocked” by different gates?

Yes, we applied the Caver tool to the whole structure, and we have done an extensive analysis of the complete structure. However, this tool should be used with caution and results should not be over-interpreted. Depending on the probe radius and starting point a much larger number of potential connections are detected, with the majority of these connections being unsuitable as ion-channels as they terminate in the membrane region or are lined with hydrophobic residues only. On the other hand, internal volumes with no connection to the exterior will not be detected by Caver. We have

focused on clear connections with a probe radius larger than or equal to the radius of the sodium ion of around 1.0 Å and made exceptions for clearly hydrophilic connections with a probe radius down to 0.8 Å.

In addition to the Phe41MrpB gate we describe the potential Phe341 gate in MrpD. We think that it blocks a proton pathway from the cytosol to the central Lys of MrpD comparable to the situation found in complex I.

We think that the model for sodium uptake by MrpD proposed by Li et al is at variance with our experimental data. We show that Glu687 in MrpA plays a central role and suggest two different sodium uptake routes converging at this residue. Neither of these uptake pathways corresponds to the Li et al. model. In addition, we have argued that sodium uptake by MrpD would mean that the positively charged ion must pass the central lysine residue of this subunit. We think that an acidic residue like Glu687 is much better suited to attract a sodium cation from the cytosolic side.

6- At what pH value were the simulations performed?

The pH was not defined explicitly in our simulations. Nevertheless, we performed MD simulations in two different states, where we defined protonation states of all titratable residues to be charged (S state) and another in which protonation state of titratable amino acids were calculated based on pKa calculations (P state). See Table S7 (updated numbering).

7- Phe341MrpD is proposed as a gate for proton transfer in MrpD. What is the gate in MrpA?

The proposed histidine switch in MrpA has an inherent gating function as it shuttles between different proton transfer pathways.

8- “Protonation state-dependent conformational dynamics of His248MrpA”.

a) Why MD simulations for the other residues with alternative conformations (Lines 115 to 141) are not described? Aren't they relevant?

We have discussed alternative conformations of other residues such as Glu687 of MrpA (See Fig. 6) and Phe41 of MrpB (See Fig. S12, updated numbering) from the perspective of MD simulations. And, yes, we think they are functionally important. As discussed in the text, the two conformations of Glu687 described may play a role in sodium transfer, whereas Phe41 may play a role in sodium gating. These residues, along with the His248 carrying segment (including backbone), showed larger conformational transitions, which were dependent on their protonation states as well as on charge states of surrounding residues. However, other residues displaying alternative conformations had relatively milder conformational changes. They are not discussed as we could not make any clear mechanistic conclusions for these other residues at this stage.

b) In the mutagenesis study, was protein presence evaluated? How can it be distinguished the presence of a non-functional protein from protein absence in these studies? How can it be distinguished a mutation reflecting stability change versus function?

We have purified the mutant Mrp antiporter from the KNabc strain used in the analysis of activity. The SDS PAGE of the purified protein proves that there are no issues with expression or stability. We have added this information to supplemental figure S11.

9- “Putative sodium uptake routes” – Did MD simulations analyse the hole structure? Again, are there other possible sodium pathways?

Yes, we analyzed the entire structure (in simulation trajectories) for potential sodium binding sites. It was rather clear for the putative sodium release route (Fig. 6D), where we saw spontaneous binding of sodium ions in protonation-state dependent fashion. However, we could not find similar spontaneous binding of sodium ions from the cytoplasmic side. Nevertheless, building upon the existing data, and the data presented in the manuscript, we proposed possible sodium uptake routes (Fig. 6). The accuracy of these predictions will indeed be a topic of future research.

Furthermore, we did not model sodium ions in MrpA and MrpD subunits, because of their strong homology to complex I subunits (ND5 and ND4/2, respectively), which have been suggested to transfer protons. Furthermore, these two subunits contain several lysine residues, and it is unclear how positively charged sodium ions will travel through paths containing lysine residues (even if neutral). In contrast, the paths discussed in Fig. 6, consist of several acidic residues that bind positively charged sodium ions and release upon protonation state changes, thereby supporting putative assignment of sodium transport route.

10- How does the proposed sodium pathways compare with equivalent pathways present in CPA1 and CPA2 members, as NhaA?

Obviously, the CPA1 and CPA2 family members have different structures as compared with Mrp antiporters. In NhaA, the binding site for protons and sodium ions consists of a small number of closely spaced residues in the center of the protein and the antiport mechanism is based on different conformations for inward and outward facing states. For the Mrp antiporter, we propose a histidine switch mechanism within a transfer pathway for protons that is spatially separated from potential sodium pathways passing the critical Glu687 in the C-terminal domain of MrpA. Coupling may occur either at the Glu/Lys pair of MrpD or in the vicinity of Glu687 of MrpA. In any case, the mechanism proposed for the Mrp-type antiporters is more complex and involves more functional elements that communicate over a substantial distance.

We point out in the introduction that the Mrp-type antiporters are quite different from the antiporters of the CPA1 and CPA2 families. A systematic analysis cannot be performed here due to space limitations and would be more appropriate for a review.

11- Discussion section:

a) What is meant by “The alternative gating of these pathways is likely to have fundamental significance for the stoichiometry of proton and sodium transport in Mrp”? What is the stoichiometry proposed?

The stoichiometry of Mrp antiporters is unknown. Li et al have suggested a stoichiometry of 4 protons per 1 sodium. Our proposed mechanism would be consistent with a stoichiometry of at least 2 protons per 1 sodium.

b) Legend of figure 7. “His248MrpA accepts the proton from Lys353MrpA and delivers it to the conserved Lys/Glu pair of the MrpA subunit...” and “Transfer of a second proton to His248MrpA (via Lys353MrpA) pushes the proton on the Lys/Glu pair of MrpA to Lys/Glu pair of MrpD by electrostatic interactions. The second proton moves energetically downhill from His248MrpA to the cytoplasm.” Why is the first proton from His248MrpA delivered to the Lys/Glu pair of the MrpA and not moved energetically downhill to the cytoplasm as the second proton?

This is an excellent question, but at this stage it cannot be answered fully. Fig. 7 provides a working hypothesis and we aim to answer this question in the future with experiments and simulations. Nevertheless, we can speculate at this stage that it is likely that the pKa of the Lys/Glu pair of MrpA is higher than His248, and as a result as soon as the histidine is protonated with the first proton, it is

transferred to the Lys/Glu pair. Moreover, we can speculate that the gate to the cytoplasmic side is likely closed when histidine receives the first proton, but opens only when the second proton arrives on histidine (together with the Lys/Glu pair protonated).

c) I missed a figure/scheme indicated the composition (amino acid side chains and respective distances) of the different proton and sodium pathways.

We appreciate the suggestion but we think such a figure would be very crowded and hard to interpret, given that there are dozens of water molecules in the central cavity and hundreds of residues surrounding them. Indeed, there are papers presenting such figures with distances between all of the key amino acids labelled. These figures only serve to 'guess' where protons/waters/sodiums might bind and travel. Since our high-resolution structure already has water molecules explicitly modelled, we don't feel a need to use such indirect measures for discussing the pathways. In addition, our observations suggest that conformational changes are involved in proton/ion translocation, making it difficult to discuss the path only from interatomic distances seen in a static structure. Therefore, we present in Fig. 7 a conceptual drawing of possible proton and sodium pathways, with key amino acids mentioned in the legends. In addition, we already present in Figs. 2C,D,E and S6 and S7 some of the key residues and distances.

12- How do the authors think the antiporter activity is activated? How is it regulated? What subunits have these roles? Any clues in the structure for these most biological relevant aspects?

In this study, we focus on the structure and the mechanism. We agree that regulation is an important question but it is too early to speculate about this. We will address this question in future experiments.

13- What is the reason for the chosen membrane composition (80% POPE and 20% POPG) in the simulations? What is the lipid composition of *Bacillus pseudofirmus*' membranes? And for 150 mM NaCl?

*Our protein samples were prepared in *E. coli* strain C41, which has PE and PG lipids in the environment, in which structural analysis was also done. Furthermore, lipid composition in *B. pseudofirmus* has major components; PG, PE and diphosphatidylglycerol. Taking into account both these aspects and to make our simulations as close as possible to experiments, we chose 80% PE and 20% PG.*

Similarly, 150 mM NaCl ion concentration (which is also used regularly in MD simulations) matches the values used in experiments (see materials and methods).

Smaller comments

Mrps are transporters (antiporters) no enzymatic activity is known to be performed.

Line 113 – catalytic cycle, should read transport cycle

Line 259 – enzymatic function, should read transport function

Line 265 – loss of enzymatic activity, should read transport activity

Line 331 – catalytic cycle, should read transport cycle

Line 106 – remove “our”

Line 109 – introduce “sp” in “Dietzia antiporter”

MD calculation – please refer to S and P as systems or states not sometimes systems and other times states.

Thank you for these comments. We have made the requested changes.

Reviewer #4 (Remarks to the Author):

This manuscript presents the third structure of an Mrp proton/sodium antiporter, in this case from *Bacillus pseudofirmus*. These antiporters or part of a superfamily of related primary and secondary ion transporters which includes the respiratory Complex I. Hence, information from the "multiple resistance and pH adaptation (Mrp) cation/proton antiporters are of interest to a wide community of researchers. What distinguishes the work presented in this manuscript is that the structure, determined by cryo-electron microscopy is at significantly higher resolution than the other structures of Mrp-type antiporters. The 2.2 Å structure reported here allows the authors to (1) plausibly identify more water molecules internally within the structure; and (2) identify multiple conformational states of amino acid side chains, portions of the polypeptide, and internal water molecules. The additional water molecules include about 70 within putative ion transport pathways. This additional structural detail is tied into MD studies which provide the basis for a model of the sequence of steps in the transporter mechanistic cycle.

The current work is very well done and clearly presented. The additional definition of possible ion transport pathways, including notably the input pathway for Na⁺, adds significantly to the literature in this field. The alternative conformations, most noteworthy being the "histidine switch" is also a significant contribution and will be of interest to others.

We thank the referee for comments.

Several points should be noted by the authors.

1. It is suggested that in Complex I, the idea of four complete and self-contained proton pumping pathways has been questioned by the authors and others, and work on the Mrp complexes will help address this question. The citations to those questioning the four pathways in Complex I do not include the authors. What is the status of this question and does the current work contribute to resolution of this question?

*Sorry, but there must be a misunderstanding. The sentence reads "With respect to complex I, we and others have recently questioned the previous consensus model that assumes four complete and self-contained proton pumping paths^{21,22}." Ref 22 is our publication Parey, K. et al. High-resolution structure and dynamics of mitochondrial complex I-Insights into the proton pumping mechanism. *Sci Adv* 7, eabj3221, doi:10.1126/sciadv.abj3221 (2021).*

While there is a clear connection to the periplasm in MrpA involving the conserved Lys in TMH12 there is no corresponding connection involving the conserved TMH12 Lys in MrpD. This means we observe a similar situation in the Mrp antiporter and in complex I. We have not excluded that a connection forms as a result of structural changes but we have no evidence for this. A comparison of the antiporter and complex I shows that although core folds are conserved, MrpA/ND5 and MrpD/ND2/ND4 are clearly different.

2. The authors make a point distinguishing between electrostatics and the protonation states of critical residues being important to the mechanism of transport. It seems that this is largely a matter of semantics since electrostatics controls protonation states. Please clarify.

Yes, we agree that both (electrostatics and protonation) are linked. One changes and the other responds. We see effects in both directions. For instance, protonation of His248 (MrpA) can change the protonation state of the KE pair of MrpA (Fig. 7) due to electrostatic interactions. Thus, here the protonation state of one residue (His248) affects the protonation state of the other residue (KE pair) by electrostatic interactions.

3. The authors point out the differences in whether neutral His248(MrpA) is protonated in the delta or epsilon position. If they believe this is important, perhaps a discussion of what factors might shift the preference for the protonation position.

Yes, our molecular dynamics simulations reveal differential dynamics of neutral His248 (MrpA) depending on where the proton is (epsilon or delta nitrogen). The protonation state change from delta to epsilon (and vice versa) or to a protonated His248 may depend on the state before and after proton delivery to and from His248. We have extended the discussion accordingly (page 12):

“The preferred conformation of His248^{MrpA} depends not only on its protonation state but also on the location of the hydrogen atom on the imidazole moiety. We propose that the ϵ nitrogen is protonated when His248^{MrpA} is in the B conformation. After protonation of the δ nitrogen via Lys353^{MrpA}, the proton from the ϵ position is delivered to the Lys/Glu pair of MrpA. His248^{MrpA} now carries a proton at the δ nitrogen and changes into the A conformation. In this state, it receives second proton from the periplasmic side via Lys353^{MrpA} and becomes charged (+1e). This drives proton transfer from the Lys/Glu pair of MrpA due to electrostatic repulsion, and the proton on the δ nitrogen of His248^{MrpA} is released to the N side. Neutral His248^{MrpA} with its ϵ nitrogen protonated reverts back to B conformation by hydrogen bonding to Ser146^{MrpA}. The protonation-state-dependent conformational dynamics of His248^{MrpA} and its connectivity to hydrated and conserved proton transfer pathways are in agreement with earlier MD simulations on bacterial and mitochondrial complex I^{22,27}.”

4. The authors suggest that it is a new observation that polar residues, in addition to ionizable residues, are important to stabilize the internal waters. Such a role for polar residues seems obvious, so why is this new?

The stabilization of internal waters by polar residues is not meant to be a new principle but our structure shows a number of protein-water contacts that have not been described before. It seemed remarkable to us that quite a few of the polar residues besides titratable ones involved are also conserved.

Overall, this is an excellent paper and worthy of publication in this journal

We very much appreciate this favorable comment.

REVIEWERS' COMMENTS

Reviewer #1 (Remarks to the Author):

The authors edited the manuscript and replied to most reviewer comments.

For my comments, the authors did not respond to the suggestion to increase the range of resolution to 2-5 Å, or at least 2.0 - 4.0 Å. A quick response to keep it that way or to edit it would be appreciated.

The authors decided to just remove indications for membrane borders, which I do not recommend. I'd like to see membrane borders in figures 1B, 2A, 3A if possible, 4A, 5 B-D if possible, S2G, S5 A and C, S6A-B, S9A and F, or wherever there is a side view shown of the protein complex.

I added a commented file to the rebuttal focussed on my comments (reviewer 1 comments).

Reviewer #2 (Remarks to the Author):

My queries are well addressed, and due diligence is offered to all the concerns. So, I am happy to accept the revised piece of work.

Reviewer #3 (Remarks to the Author):

As mentioned in my previous assessment, the work of Lee et al, describing the cryo-EM structure and molecular dynamics simulations on the Mrp-type antiporter from *Bacillus pseudofirmus*, is very interesting and pertinent.

In the revised version of the manuscript the authors addressed most of my concerns, however I still feel the need to contextualize the discussion of the transport mechanism (my previous point 12): what do the authors hypothesize the trigger of the transport is and, considering that this antiport is present in an alkalophile, what the driving force for the transport is.

The Mrp antiporter discussed in the article is from *Bacillus pseudofirmus*, an alkalophilic bacterium that lives at pH between 9-11, being able to maintain its internal pH two units lower than the external pH (this means a higher proton concentration inside than outside of the cell). Nevertheless, *Bacillus pseudofirmus* establishes a membrane potential of -180 mV (more positive outside than inside, possible due to differences in sodium concentration, which is higher outside).

If the authors introduce in figure 7 i) the polarity of the membrane (minus (-) to the cytoplasm and plus

(+) periplasm), ii) pH values (around 10.5 outside and 8.2 inside) and iii) [Na⁺] values (higher outside), how would the proposed mechanism take place? In this case, how would the membrane potential be dissipated since the ions (H⁺ and Na⁺) would be transported against their respective concentration differences?

(Please correct the stoichiometry in figure 7, there should be 2/3 protons being uptake so 2/3 protons can be delivered to the other side).

Reviewer #4 (Remarks to the Author):

The revised version of the manuscript has addressed the major points of criticism and I am satisfied.

Point-by-point response to the reviewers' comments

Reviewer #1 (Remarks to the Author):

The authors edited the manuscript and replied to most reviewer comments.

For my comments, the authors did not respond to the suggestion to increase the range of resolution to 2-5 Å, or at least 2.0 - 4.0 Å. A quick response to keep it that way or to edit it would be appreciated.

The authors decided to just remove indications for membrane borders, which I do not recommend. I'd like to see membrane borders in figures 1B, 2A, 3A if possible, 4A, 5 B-D if possible, S2G, S5 A and C, S6A-B, S9A and F, or wherever there is a side view shown of the protein complex.

I added a commented file to the rebuttal focussed on my comments (reviewer 1 comments).

We have added a new panel to Supplementary Fig. S2 that shows the resolution range between 2.0 and 4.0 Å. We have also added membrane boundaries to figures which show a side view of the antiporter complex.

Reviewer #3 (Remarks to the Author):

As mentioned in my previous assessment, the work of Lee et al, describing the cryo-EM structure and molecular dynamics simulations on the Mrp-type antiporter from *Bacillus pseudofirmus*, is very interesting and pertinent.

In the revised version of the manuscript the authors addressed most of my concerns, however I still feel the need to contextualize the discussion of the transport mechanism (my previous point 12): what do the authors hypothesize the trigger of the transport is and, considering that this antiport is present in an alkalophile, what the driving force for the transport is.

The Mrp antiporter discussed in the article is from *Bacillus pseudofirmus*, an alkalophilic bacterium that lives at pH between 9-11, being able to maintain its internal pH two units lower than the external pH (this means a higher proton concentration inside than outside of the cell). Nevertheless, *Bacillus pseudofirmus* establishes a membrane potential of -180 mV (more positive outside than inside, possibly due to differences in sodium concentration, which is higher outside).

If the authors introduce in figure 7 i) the polarity of the membrane (minus (-) to the cytoplasm and plus (+) periplasm), ii) pH values (around 10.5 outside and 8.2 inside) and iii) [Na⁺] values (higher outside), how would the proposed mechanism take place? In this case, how would the membrane potential be dissipated since the ions (H⁺ and Na⁺) would be transported against their respective concentration differences?

(Please correct the stoichiometry in figure 7, there should be 2/3 protons being uptake so 2/3 protons can be delivered to the other side).

We thank the reviewer for this important note. Under alkaline conditions the Mrp antiporter is essential for *B. pseudofirmus* to stabilize its cytosolic pH. In fact, alkaline conditions are a challenge for chemiosmotic coupling because the protons have to move against a concentration gradient. This is only possible because the membrane potential is very negative (-180 mV). The driving force for inward proton transfer is thus purely electrical in contrast to the standard

OXPHOS systems such as mitochondria. The membrane potential in B. pseudofirmus has been determined experimentally and responds to environmental conditions. It is also dependent on the sodium cycle and requires that the exterior sodium concentration is considerably higher than in the cytosol. The antiporter plays an important role for sodium extrusion. However, other systems may also contribute to the sodium gradient. We think that this is an unresolved issue in the physiology of B. pseudofirmus and prefer not to speculate about it here. The important point is that the Mrp antiporter can only function when the stoichiometry results in an electrogenic transfer of protons and sodium with the number of transferred protons exceeding the number of transferred sodium ions. To make this point clear, we have added the following paragraph at the beginning of the discussion section.

“Under alkaline conditions, B. pseudofirmus is dependent on the activity of its Mrp antiporter that allows to stabilize the internal pH value about two units below that of the external environment ^{1,3,23}. Accordingly, the transfer of protons inward occurs against a concentration gradient and is possible only because the driving force is provided by the electrical component of the membrane potential. The membrane potential of ~ -180 mV (inside negative) is based on the balance between a proton cycle and a sodium cycle. The stoichiometry of the Mrp type antiporters is unknown but is predicted to be electrogenic with the number of transferred protons exceeding the number of transferred sodium ions ^{1,3,14,15}.”

We also clarified this point in the legend of Figure 7. There we have also made it clear that the dotted lines are alternative routes for proton transfer. Our scheme supports a stoichiometry of 2 H⁺/1 Na⁺. However, we would like to stress that at this stage the stoichiometry is unknown and could be higher, as suggested in ref. 15.